

# Evaporative controls on Antarctic precipitation: An ECHAM6 model study using novel water tracer diagnostics

Qinggang Gao[1,2], Louise C. Sime[1], Alison McLaren[1], Thomas J. Bracegirdle[4], Emilie Capron[5], Rachael H. Rhodes[2], Hans Christian Steen-Larsen[6], Xiaoxu Shi[3], and Martin Werner[3]

[1]Ice Dynamics and Paleoclimate, British Antarctic Survey, Cambridge, United Kingdom
[2]Department of Earth Sciences, University of Cambridge, Cambridge, United Kingdom
[3]Alfred Wegener Institute, Helmholtz Centre for Polar and Marine Research, Bremerhaven, Germany
[4]Atmosphere, Ice and Climate, British Antarctic Survey, Cambridge, United Kingdom
[5]Université Grenoble Alpes, CNRS, IRD, Grenoble INP, IGE, Grenoble, France
[6]Geophysical Institute, University of Bergen and Bjerknes Centre for Climate Research, Bergen, Norway

**Correspondence:** Qinggang Gao (qino@bas.ac.uk)

**Abstract.**

Improving our understanding of the controls on Antarctic precipitation is critical for gaining insights into polar, and global changes. Here we develop and implement innovative water tracing diagnostics in the atmospheric general circulation model ECHAM6. These tracers provide new precise information on moisture source locations and properties of Antarctic precipita-

tion. In our preindustrial simulation, annual mean Antarctic precipitation originating from the open ocean has a source latitude range of 49-35° S; a source sea surface temperature range of 9.8-16.3°C; a source 2 m relative humidity range of 75.6-83.3%; and a source 10 m wind speed (wind10) range of 10.1 to 11.3 m s$^{-1}$. The tendency of poleward vapour transport to follow moist isentropes means that central Antarctic precipitation is sourced from more equatorward (distant) sources via elevated transport pathways than coastal Antarctic precipitation. We find however this tendency breaks down in the lower troposphere,

likely due to diabatic cooling. Heavy precipitation is sourced by longer-range moisture transport: it comes from 2.9° (300 km, averaged over Antarctica) more equatorward (distant) sources compared to the rest of precipitation. Precipitation during negative phases of the Southern Annular Mode (SAM) also comes from more equatorward moisture sources (by 2.4°, averaged over Antarctica) than precipitation during positive SAM phases, likely due to amplified planetary waves during negative SAM phases. Moreover, source wind10 of annual mean precipitation is on average 2.1 m s$^{-1}$ higher than annual mean wind10 at

the evaporation source locations from which the precipitation originates. This shows that the evaporation of moisture driving Antarctic precipitation occurs under windier conditions than average. This is the first time this particular thermodynamic control of Southern Ocean surface wind on moisture availability for Antarctic precipitation has been quantified. Overall, our novel water tracing diagnostics enhance our understanding of the controlling factors of Antarctic precipitation.

## 1 Introduction

Antarctic climate is changing. The years 2022 and 2023 both witnessed new minima in sea ice extent and some of the largest extreme heat and precipitation events. Increased moisture in Antarctic regions can directly drive warming: a range of model





simulations show that increased poleward moisture transport in a warmer world is the largest contributor to Antarctic warming (Hahn et al., 2021). On top of this, the warming ocean around Antarctica is very likely to lead to ice mass loss via sub-shelf melting and calving (DeConto and Pollard, 2016; DeConto et al., 2021). These Southern Ocean changes may impact local
evaporation to drive Antarctic precipitation changes and therefore influence the Antarctic surface mass balance (Mottram et al., 2021; Lenaerts et al., 2019). It is possible that under a warmer future increases in Antarctic vapour and precipitation may contribute to changes in surface mass balance and extreme warming episodes (Davison et al., 2023; Medley and Thomas, 2018; Frieler et al., 2015; Winkelmann et al., 2012). Overall, projections of Antarctic contribution to future sea level rise due to these surface mass balance processes remain uncertain (IPCC, 2022).

Antarctic precipitation frequently falls as near-continuous clear-sky precipitation, so-called diamond dust (Bromwich, 1988). However, there are also relatively short-lived intrusions of maritime air, which can lead to episodes of heavier precipitation (Turner et al., 2019). Indeed, these events may contribute to 30-70% of total precipitation across Antarctica, with likely more than 30% of precipitation in the interior, and up to 70% in coastal regions (Turner et al., 2019). Whilst the mass balance of Antarctica can be estimated from satellite altimetry, gravimetry, and interferometry measurements (The IMBIE team, 2018),
we still know surprisingly little about thermodynamic and dynamic drivers of Antarctic precipitation.

Marine air intrusions (Schlosser et al., 2010), sometimes in the form of atmospheric rivers (Gorodetskaya et al., 2014; Wille et al., 2021), tend to occur during periods of strong meridional flow (Noone et al., 1999; Adusumilli et al., 2021). This is conducive for the advection of moist and warm air from relatively low latitudes (Schlosser et al., 2010; Dittmann et al., 2016). These conditions can occur alongside periods of planetary wave amplification (Hirasawa et al., 2000; Massom et al., 2004).
Indeed, persistent ridges or dipolar patterns (with high pressure to the east and low to the west) are known to have contributed to heavy precipitation events across a range of Antarctic sites, including EPICA Dome Concordia (EDC, Schlosser et al., 2016); Dome Fuji (Dittmann et al., 2016); and Dronning Maud Land (Gorodetskaya et al., 2014; Terpstra et al., 2021; Kurita et al., 2016; Noone et al., 1999). Thus while these marine air intrusions are mainly known for heavy precipitation events at coastal locations, they also play a major role in heavier precipitation events in the interior of Antarctica (Genthon et al., 1998;
Gorodetskaya et al., 2014). The identification of source properties associated with these events is useful for predicting the evolution of precipitation across the whole of Antarctica under global and polar warming.

Compared to heavy precipitation events, light precipitation events such as diamond dust seem to have received less attention. However, dependent on the definitions used, light precipitation may dominate total precipitation over inland Antarctica (Stenni et al., 2016); and similar to heavy precipitation, light precipitation also depends on synoptic conditions (Schlosser et al., 2010).
Developing an improved understanding of drivers of light precipitation is thus also important.

Variations in Antarctic precipitation have been linked to the principal modes of atmospheric circulation variability at southern high latitudes, particularly SAM and Pacific-South American patterns associated with El Niño–Southern Oscillation (Marshall et al., 2017). The variations are associated with changes in the zonal and meridional flows of atmospheric moisture around and towards Antarctica. While positive SAM polarity is linked to increased cyclogenesis and poleward storm track migration
(Uotila et al., 2013; Fogt and Marshall, 2020), Antarctic regions do not show a uniform relationship between SAM and precip-



itation (Marshall et al., 2017; Medley and Thomas, 2018). It is not yet clear how SAM variations, and associated changes in moisture flux tracks, will impact precipitation across Antarctica.

Insights into Antarctic precipitation can be gleaned through its evaporative source regions and properties obtained from modelling studies. The most commonly applied tool for this is backward trajectory models (Sodemann and Stohl, 2009; Gimeno
et al., 2010). Of backward trajectory studies, results regarding Antarctic precipitation sources from Sodemann and Stohl (2009) are probably more reliable than other Lagrangian studies with shorter (usually five-day) backward trajectories (*e.g.* Gimeno et al., 2010). However, Sodemann and Stohl (2009) could still only attribute ∼90% of total precipitation to specific sources and noted issues with the identification of precipitation events solely using thresholds in specific humidity changes.

In addition to the Lagrangian trajectory approach, general circulation models (GCMs) can be equipped with water tracers
to identify moisture sources. The water tracers track moisture that is evaporated from prescribed regions until it precipitates (*e.g.* Koster et al., 1986, 1992; Delaygue et al., 2000; Werner et al., 2001; Singh et al., 2016; Wang et al., 2020). Typically, the globe is divided into multiple source regions, and then the contribution of each region to total precipitation at any location can be quantified. For example, Koster et al. (1992) used water tracers to show that the average evaporative source sea surface temperature (SST) of Antarctic July precipitation is around 11.6°C. While Bintanja and Selten (2014) quantified contributions
of local and remote sources to Arctic precipitation through budget methods, this type of source attribution was shown to be biased by Singh et al. (2017) using water tracers.

Recently, Fiorella et al. (2021) introduced a new approach to using water tracers in GCMs. Their process-oriented water tracers can track moisture properties related to evaporation, transport, and condensation. This approach is more computationally efficient than the previous approach of tagging moisture from individual predefined regions, and it prevents biases while
estimating evaporative source properties (see Appendix A for details). Here, we employ and further develop this approach to enable us, for the first time, to precisely quantify evaporative source locations and properties of Antarctic precipitation.

The manuscript is organised as follows: Section 2 introduces the materials and methods, and Section 3 presents the results; conclusions and perspectives are given in Section 4.

## 2   Materials and methods

The atmospheric GCM ECHAM6 and our simulation setup are described in Section 2.1. Then Section 2.2 presents our development of new water tracer methods within the GCM. Revised definitions of Heavy Precipitation (HP) and Light Precipitation (LP) are given in Section 2.3, followed by the definition of SAM used in this study in Section 2.4.

### 2.1   Model and simulation

For this study, we use the ECHAM6 atmospheric GCM, which was developed by the Max-Planck-Institute for Meteorology
(MPI-M) in Hamburg (Stevens et al., 2013). In ECHAM6, the primitive equations are formulated in a mixed finite-difference and spectral discretisation with a semi-implicit time scheme. The dynamical part is represented by truncated series of spherical harmonics in the horizontal and a finite-difference scheme in the vertical. Moisture transport is treated using a mass-conserving





flux form semi-Lagrangian algorithm on a Gaussian grid. The vertical coordinate consists of a hybrid sigma-pressure coordinate system, which is terrain-following at lower levels and flattens to surfaces of constant pressure at upper levels. We use a T63L47

resolution, *i.e.* a resolution equivalent to $1.87° \times 1.87°$ horizontal grid size and 47 vertical levels extending to 0.01 hPa. This resolution captures the overall shape of the Antarctic ice sheet with the caveat that the interior of the East Antarctic Ice Sheet (EAIS) may be slightly too low in elevation, and complex coastal topography is not captured well (Fig. B1).

We set up a preindustrial condition simulation using sea surface temperature (SST) and sea ice concentration (SIC) data from the Atmospheric Modelling Intercomparison Project (AMIP, Fig. B2a and B2b). These are climatological monthly mean

data from 1870 to 1899 (Durack et al., 2022). For sea ice-covered areas, SST is set to $-1.8°$C. We run our simulation for 60 years and use the last 50 years in the analysis. Daily ECHAM6 model output is used for our analyses.

The formulation of air-sea moisture fluxes in the model is relevant for moisture source properties. In ECHAM6, oceanic evaporation is estimated based on bulk parameterisation (Hoffmann et al., 1998; Liu et al., 1979; Yu and Weller, 2007):

$$E = \rho C_e |V| (q_{sat}^{sfc} - q^{near\_sfc}), \tag{1}$$

where $E$ represents evaporation, $\rho$ the air density, $C_e$ the turbulent exchange coefficient related to atmospheric stability (Fairall et al., 2003), $|V|$ the difference in wind speed between the lowest model level and the surface, $q_{sat}^{sfc}$ the saturation specific humidity at the surface, and $q^{near\_sfc}$ the specific humidity above the surface.

## 2.2 Water tracing methods

Previous versions of ECHAM had both water isotopes and standard water tracers incorporated (Hoffmann et al., 1998; Werner

et al., 2001). However, the latest version of ECHAM, ECHAM6, has so far only been equipped with water isotope tracers (Cauquoin et al., 2019). Building upon the model code infrastructure of water isotopes, for this work we implemented two types of water tracers: 1) standard water tracers, which are usually applied to track water evaporating from prescribed regions; and b) scaled-flux water tracers, which follow the concepts of Fiorella et al. (2021) and were referred to as process-oriented tracers in their paper. These two tracer sets are used together here in a new and complementary approach.

For the standard water tracers (hereafter "prescribed-region" tracers), we prescribe seven complementary regions. These are: the open ocean south of 50° S; Southern Hemisphere (SH) sea ice; Pacific, Indian, and Atlantic oceans north of 50° S; the Antarctic ice sheet (AIS); and land exclusive of AIS. As sea ice changes at each time step, the prescribed SH sea ice region follows the changes. This is in itself a new form of dynamic prescribed-region water tracing. Where a grid cell contains both open ocean and sea ice, we track these sub-grid-scale fluxes separately.

The implementation of scaled-flux water tracers follows Fiorella et al. (2021), with some modifications for ECHAM6 as described in Appendix A. The scaled-flux tracing method can be used to tag any property associated with evaporation. Given recent interests in how the changing Southern Ocean will affect Antarctic precipitation, we focus here on properties which are most closely associated with evaporation. Based on Eq. 1, in ECHAM6 $|V|$ is approximated as 10 m wind speed (wind10); $q_{sat}^{sfc}$ depends on SST; and $q^{near\_sfc}$ is approximated as 2 m specific humidity, which is linked to 2 m relative humidity (rh2m)

and associated air temperature (Yu and Weller, 2007). So, we chose to trace source longitude, latitude, SST, rh2m, and wind10.





Please see Appendix A for a fuller description of how the scaled-flux water tracers are implemented in ECHAM6, and how the results compare with prescribed-region tracers.

Based on the source latitude and longitude of precipitation and precipitation site location, a source-sink distance can be estimated by calculating the geographical distance from moisture source to precipitation site assuming a spherical earth sur-
face. Note that this calculated geographical distance is not the same as the actual transport distance of the moisture parcel. Nevertheless, this source-sink distance is physically meaningful and is likely very closely associated with the actual modelled moisture transport distance.

### 2.3 Defining Heavy and Light Precipitation

The identification of source properties associated with Light and Heavy Precipitation (LP and HP) is useful for understanding
precipitation drivers associated with recent Southern Ocean changes. Here we lay out the definitions used herein for LP and HP. This first requires us to define a "precipitation day" in Antarctica.

Turner et al. (2019) defined a precipitation day in Antarctica to have more than $0.02 \ \mathrm{mm \ day^{-1}}$ precipitation. However, a threshold of $0.02 \ \mathrm{mm \ day^{-1}}$ excludes low daily precipitation amounts that can contribute to more than $10\%$ of total precipitation amount over the Antarctic interior in both the ERA5 reanalysis (Hersbach et al., 2020) and our simulation. We therefore
use a lower threshold of $0.002 \ \mathrm{mm \ day^{-1}}$. This ensures we account for more than $99.7\%$ of the total precipitation amount at every Antarctic grid cell.

We define LP as that which cumulatively contributes to $10\%$ of total precipitation, while all other precipitation days have higher precipitation rates. For the definition of HP, we follow the Turner et al. (2019) definition and use the top $10\%$ precipitation days. Note the definition of LP, based on precipitation amount, is different from that of HP, based on precipitation rates.
This is because a definition of LP as the $10\%$ lowest precipitation days would contribute to only $0.3\%$ of Antarctic precipitation. This difference is therefore to ensure that our LP results are robust, whilst still using a recognised definition of HP (*c.f.* Turner et al., 2019).

### 2.4 The Southern Annular Mode (SAM) index

We calculate monthly SAM values as the difference in normalised zonal mean sea level pressure at $40°$ S and $65°$ S (Gong and
Wang, 1999). We define SAM+ and SAM- months as months with SAM values deviating more than one standard deviation from the mean (calculated from the 50-year period) in the positive and negative directions, respectively.

## 3 Results

This section starts with an evaluation of the ECHAM6 model performance in reproducing reconstructed Antarctic accumulation and the SAM (Section 3.1). This evaluation ensures we have confidence in the application of the simulation to investigate
source regions (Section 3.2) and source properties (Section 3.3) of Antarctic precipitation. Moisture sources of HP and LP are investigated in Section 3.4. Relationships between SAM and the evaporative sources of precipitation are explored in Section 3.5.



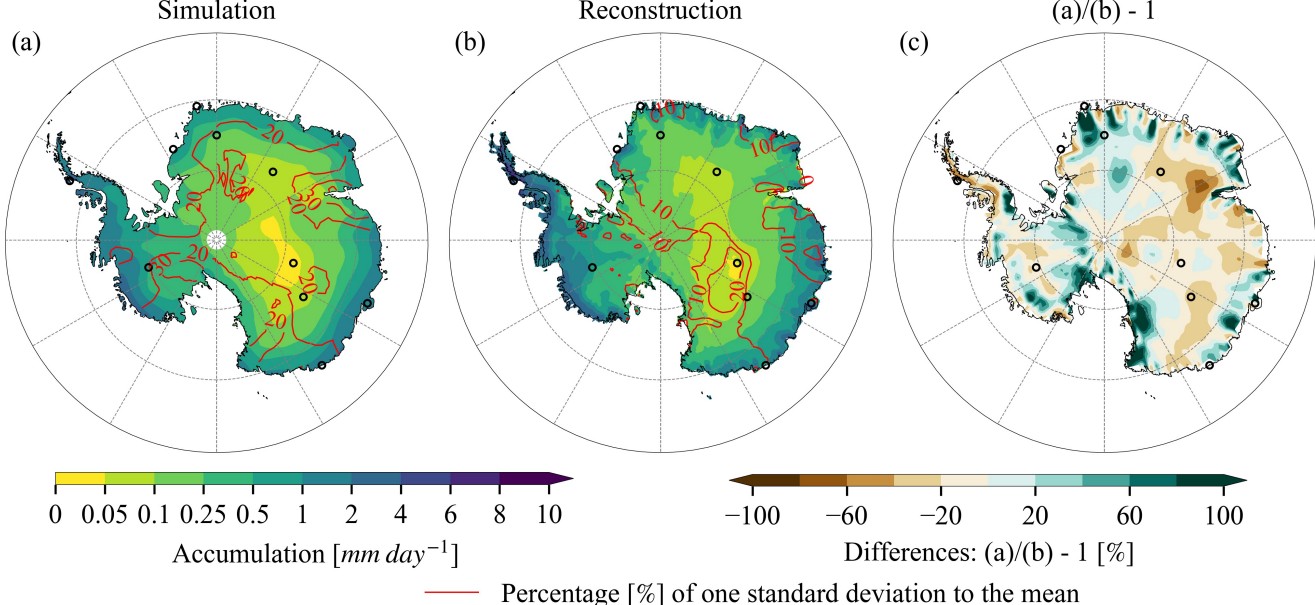

**Figure 1.** Annual mean accumulation rate over Antarctica in (a) our preindustrial simulation, and (b) the reconstruction of Medley and Thomas (2018) for the period 1800-1900. The Medley and Thomas (2018) dataset is based on combining ice core data with spatial patterns of accumulation derived from the MERRA-2 reanalysis (Gelaro et al., 2017). (c) Differences as a percentage of the Medley and Thomas (2018) reconstruction. For the comparison, both datasets are regridded to $1° \times 1°$ grids using a bilinear method. Accumulation in the simulation is defined as differences between precipitation and evaporation, while post-depositional effects are not considered. Black empty circles represent ten sites whose names are given in Fig. B1b.

## 3.1 The simulation of Antarctic precipitation and the SAM

The overall spatial patterns of accumulation (precipitation minus evaporation) are captured in the ECHAM6 model results (Fig. 1). Though it is on the low side over the Antarctic plateau and the Antarctic Peninsula (AP) compared to the Medley and

Thomas (2018) reconstruction, and is high across some coastal areas. This could be partly due to the relatively coarse (T63) spatial resolution of our simulation, though it could also be related to uncertainties that afflict all reconstructions of Antarctic accumulation (Monaghan et al., 2006). Interannual variability, measured as the percentage of annual standard deviation to the annual mean, is in the same order for both datasets: ~20% for the ECHAM6 simulation and ~10% for the Medley and Thomas (2018) dataset.

The annual cycle of Antarctic precipitation in our simulation is similar to ERA5 (Fig. B3), with precipitation averaged over Antarctica exceeding $15 \mathrm{~mm~month^{-1}}$ from March to August; a peak in May; and a minimum in December-January. Spatial patterns of HP contributions to total precipitation in our simulation are likewise very similar to those in Turner et al. (2019), with high values around major ice shelves (Fig. B4).





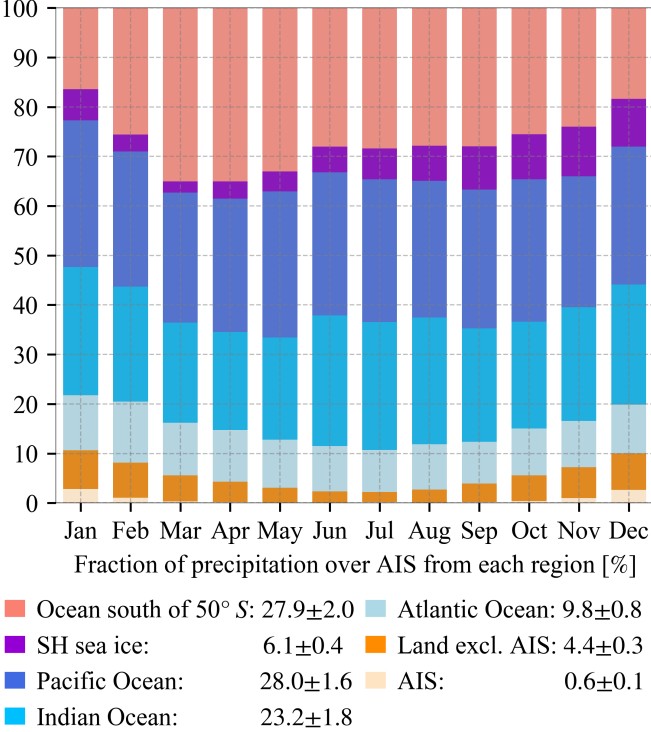

**Figure 2.** Relative contributions of seven prescribed regions to monthly mean precipitation integrated over Antarctica. The contributions (± one standard deviation) to annual mean precipitation over Antarctica are given in the legend.

We evaluated our modelled SAM index against the SAM index based on station observations between 1971-2000 (Marshall, 2003). Due to the SAM definition, both datasets have similar mean values and standard deviations. We therefore look at monthly zonal mean sea level pressure (MSLP) at 40° S and 65° S, and their differences, to check whether our simulation features a realistic SAM. Standard deviations and RMSE suggest that both datasets have similar statistical properties (Fig. B5).

### 3.2 Where does Antarctic precipitation come from?

We find that 89% of modelled Antarctic precipitation comes from oceanic evaporation (Fig. 2 and Fig. 3a). Less than 1% of the precipitation is sourced from continental sublimation over Antarctica (Fig. 2 and Fig. 3c). The continental recycling occurs mainly around major ice shelves in December and January (up to 3%) with the most intense solar insolation. Antarctic precipitation sourced from other land masses is higher (by ~4%) than that from Antarctica itself. Similar to the CESM1 simulation of Wang et al. (2020), in our ECHAM6 simulation most of the non-Antarctica land-sourced precipitation arrives in austral summer (contributing to 8% of summer precipitation, compared to only 2% of winter precipitation). Moisture originating from these other land masses has a relatively larger contribution to EAIS precipitation (5.6%), compared to the West AIS (WAIS, 2.6%) and AP (2.7%, Fig. 3b). The remaining Antarctic precipitation (6%) is sourced from SH sea ice areas. This surface type





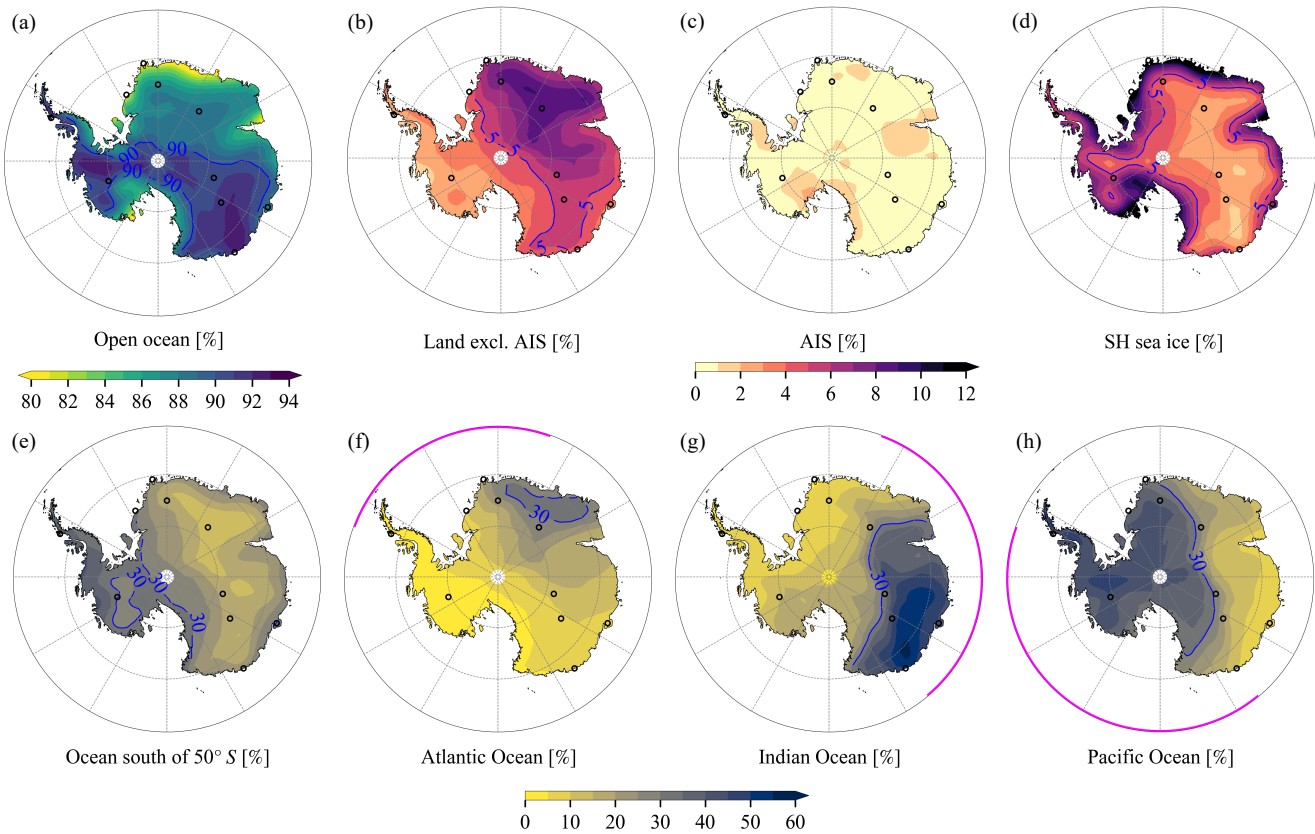

**Figure 3.** Relative contributions of prescribed regions to annual mean precipitation across Antarctica. The prescribed regions include (a) the open ocean, (b) land exclusive of AIS, (c) AIS, (d) SH sea ice, (e) the open ocean south of 50° S, (f) Atlantic, (g) Indian, and (h) Pacific ocean north of 50° S. Relative contributions from the open ocean (a) are the sum of (e)-(h). Magenta lines in (f)-(h) represent the Atlantic, Indian, and Pacific Ocean sectors, respectively. Blues lines in each figure are contours of relative contributions.

has notably larger contributions in coastal regions (Fig. 3d). Precipitation sourced from sea ice reaches the maximum between September and December (10%), due to combined influences of a relatively large sea ice area and increased solar insolation.

Regarding precipitation sourced from the open ocean, 28% of this precipitation comes from the open ocean south of 50° S.

This region contributes a larger proportion of precipitation over WAIS (35%) and AP (36%) compared to EAIS (23%, Fig. 3e). Contributions from both the Indian Ocean (23%) and the Pacific Ocean (28%) are two to three times that from the Atlantic Ocean (10%) north of 50° S. This is at least partly attributable to the sizes of these ocean basins: between the equator and 50° S, areas of the Indian and Pacific Oceans are 1.4 and 2.3 times that of the Atlantic Ocean, respectively. The three ocean basins contribute relatively more precipitation within their corresponding Antarctic sectors, though with a tendency to an eastward

shift (∼30-60°) due to the predominant eastward transport of water vapour around Antarctica (Fig. 3f-h).



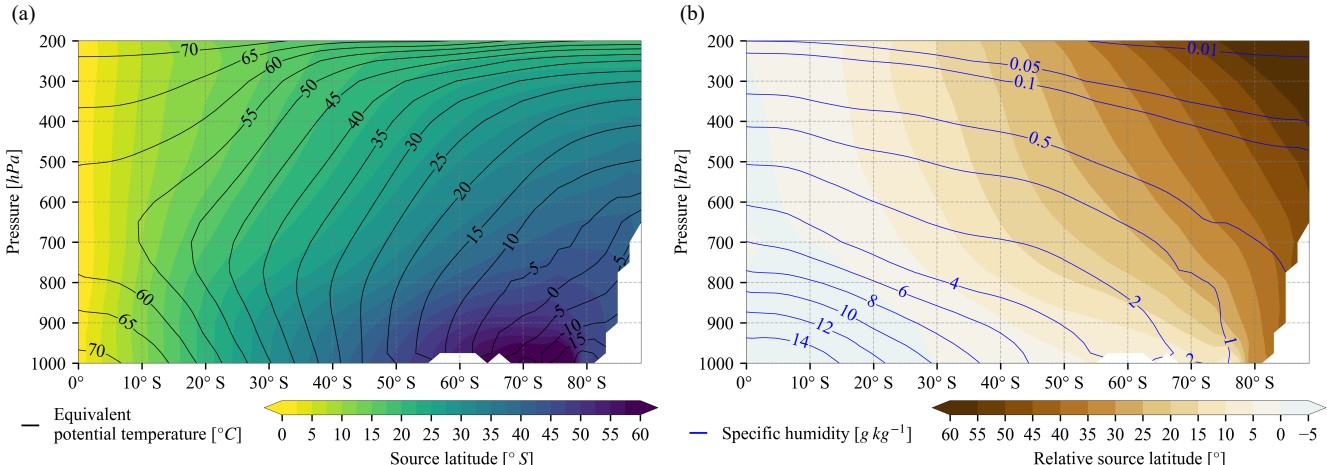

**Figure 4.** Zonal-averaged mass-weighted mean open-oceanic evaporative (a) source latitude and (b) relative source latitude of annual mean atmospheric humidity. Black contours in (a) show the zonal mean annual mean equivalent potential temperature at an interval of 5°C. Blue contours in (b) show zonal mean annual mean atmospheric specific humidity at values of [0.01, 0.05, 0.1, 0.5, 1, 2, 4, 6, 8, 10, 12, 14] g kg$^{-1}$. Relative source latitude is defined as differences between source latitude and local latitude. Positive source latitude difference means more equatorward.

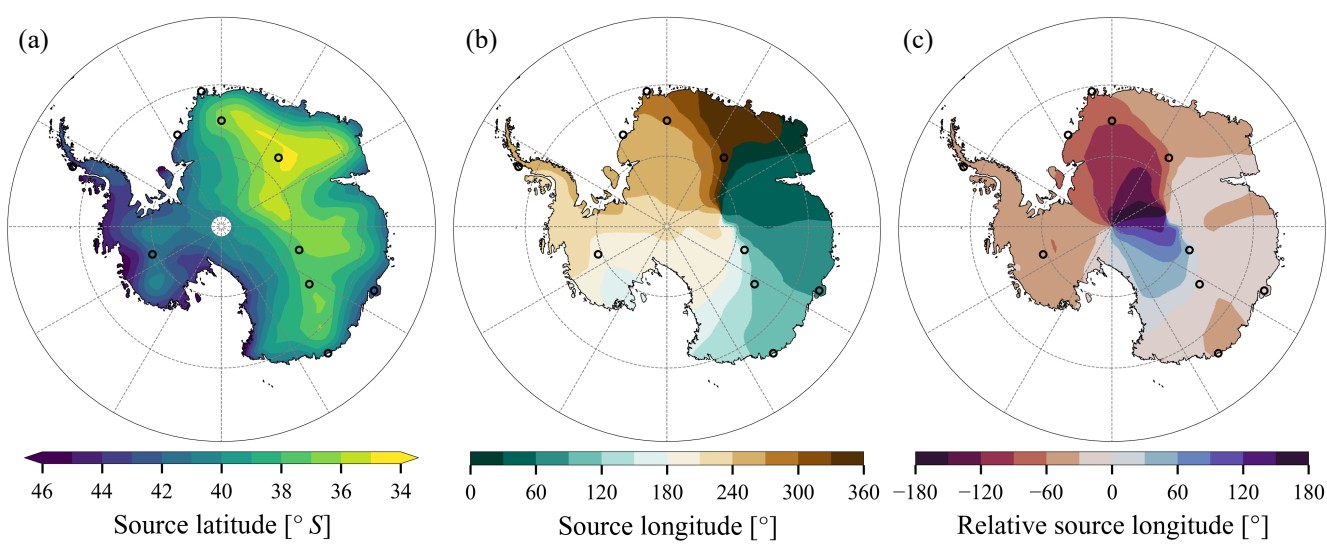

**Figure 5.** Mass-weighted mean open-oceanic evaporative (a) source latitude, (b) source longitude, and (c) relative source longitude of annual mean precipitation. Relative source longitude is estimated as differences between source longitude and local longitude. Positive source longitude difference means more eastward.



Water vapour tends to take an elevated pathway to central Antarctica (Noone and Simmonds, 2002; Wang et al., 2020). As a result, the higher, remote central regions of Antarctica tend to receive moisture sourced from more equatorward regions. Moisture sourced from more poleward ocean regions, *e.g.* south of 50° S and SH sea ice compared to land exclusive of AIS, is transported at lower altitudes to Antarctica, so its precipitation contributions are larger over WAIS and AP with lower elevations than EAIS (Fig. 3b, 3d, and 3e). At mid-to-upper troposphere, poleward moisture transport tends to follow moist isentropes, *i.e.* contours of equivalent potential temperature (Fig. 4; Pauluis et al., 2010; Bailey et al., 2019). This tendency to follow contours of equivalent potential temperature breaks down in the lower troposphere, with moisture transport pathways intersecting with moist isentropes from the surface to around 700 hPa. This breakdown may be due to diabatic cooling.

Elevated transport pathways to central Antarctic regions impact all source properties, including mass-weighted mean open-oceanic evaporative source latitude (source latitude thereafter). Source latitude of annual mean precipitation ranges from 49 to 35° S across Antarctica, and averages to 41° S over entire Antarctica (Fig. 5a). These values are close to the estimate from Sodemann and Stohl (2009) of 45 to 40° S across the Antarctic Plateau. The elevated transport pathways mean that source latitude of EAIS precipitation is more equatorward by ~3° compared to that of WAIS and AP (40° S vs. 43° S). Also, Antarctic precipitation at surface elevations above 2250 m comes from more equatorward regions by 4° compared to precipitation occurring below 2250 m (38° S vs. 42° S).

Regarding seasonality, source latitudes are most equatorward in December-January-February (DJF) and most poleward in March-April-May (MAM) and June-July-August (JJA) (Fig. B6a; an average 3.3° DJF to JJA shift over Antarctica). Reduced meridional thermal gradients and thus milder moist isentropes in DJF compared to JJA likely promote equatorward shifted moisture source regions. Weaker westerlies in DJF compared to JJA, induced by smaller meridional thermal gradients, may also play a role (see Section 3.5 for details).

Antarctic precipitation generally comes from the west (Fig. 5b-c), except for precipitation in a sector between the South Pole and EDC which appears to originate from the east. We speculate that this might be from the far west, with a rotation of more than 180 degrees, probably under impacts of the Amundsen Sea Low. Source longitude over Antarctica displays the largest inter-annual variability of all source properties (Fig. B6b).

### 3.3 What are the evaporative source properties associated with Antarctic precipitation?

Having dealt with source surface types, alongside latitude and longitude, we now consider other oceanic source properties which control evaporation: wind10, rh2m, and SST (Eq. 1).

Source SST of annual mean precipitation varies between 9.8 and 16.3°C across Antarctica, averaging to 12.8°C (Fig. 6a). This lies in the middle among estimates in the literature: 15-22°C by Petit et al. (1991), 9-14°C by Koster et al. (1992), and 10-12°C by Delaygue et al. (2000). Analogous to source latitude, EAIS precipitation originates from warmer oceans by ~1°C than WAIS and AP (13.3 vs. 12.1 and 12.4°C), and Antarctic regions at altitudes higher than 2250 m receive precipitation from warmer oceans by 2°C than lower regions (14.5 vs. 12.5°C). Source rh2m of annual mean precipitation ranges from 75.6% to 83.3% across Antarctica, and averages to 78.3% (Fig. 6b). Again, EAIS derives its precipitation from oceans with lower rh2m than WAIS and AP by 1% and 1.5%, respectively (77.9% vs. 78.9% and 79.4%), and Antarctic regions above 2250 m





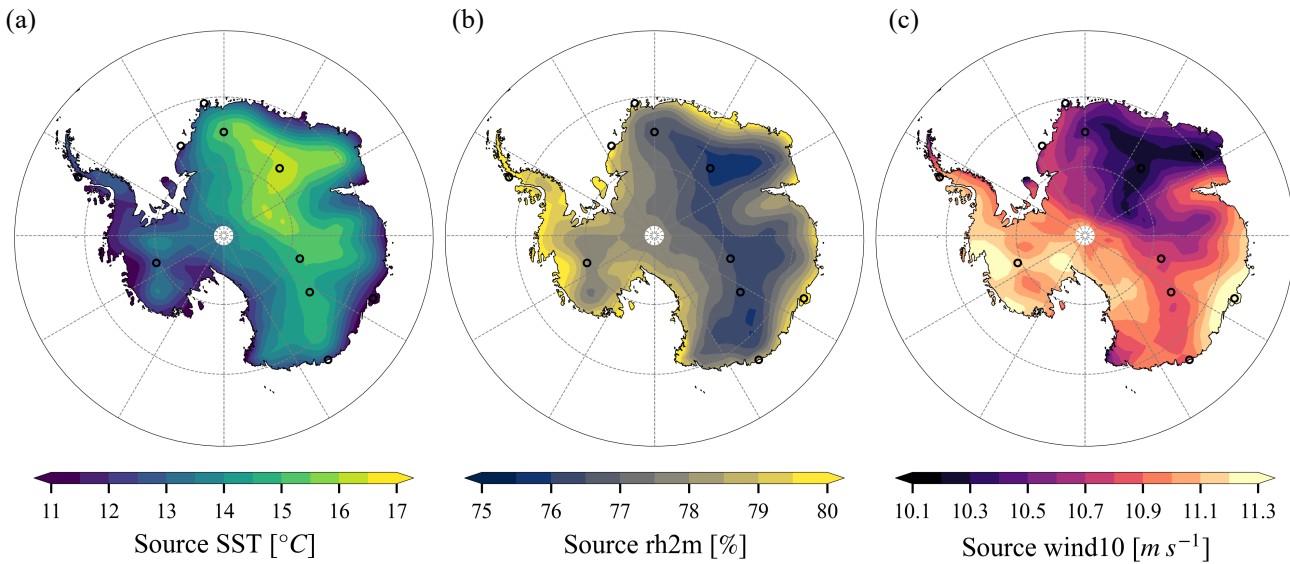

**Figure 6.** Mass-weighted mean open-oceanic evaporative (a) source SST, (b) source rh2m, and (c) source wind10 of annual mean precipitation.

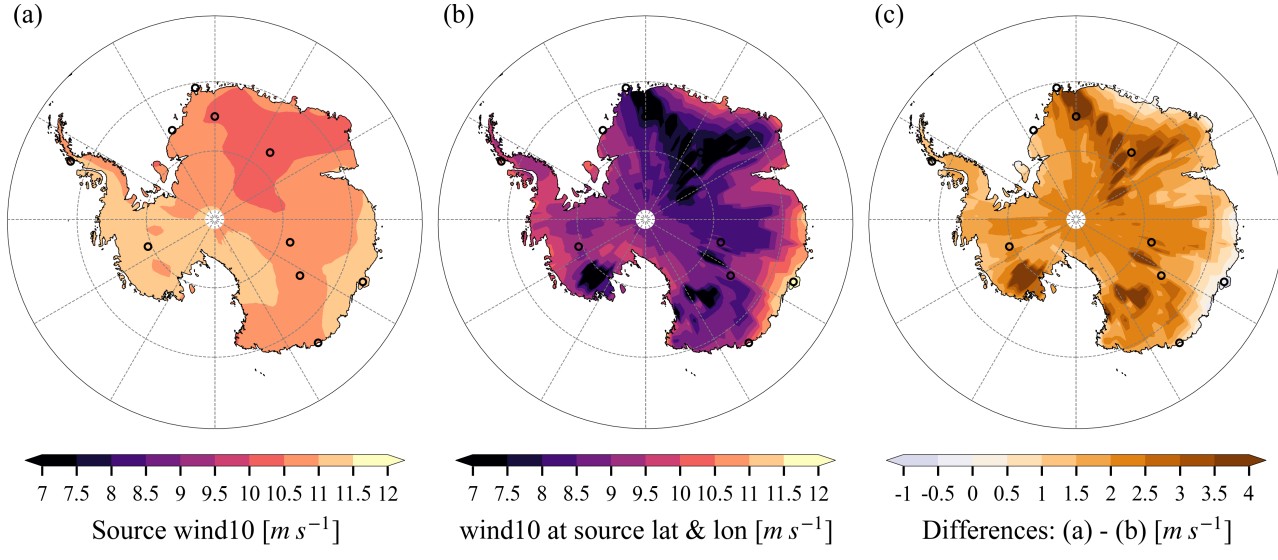

**Figure 7.** (a) Mass-weighted mean open-oceanic evaporative source wind10 of annual mean precipitation. (b) Annual mean wind10 at source locations of annual mean precipitation. (c) Differences between (a) and (b). The average difference over Antarctica is $\sim 2.1$ m s$^{-1}$. The difference indicates the impact of surface wind on evaporation which drives Antarctic precipitation.





elevations obtain precipitation from oceans with lower rh2m by 1.7% than lower regions (76.9% vs. 78.6%). Interestingly, source wind10 of annual mean precipitation has a very narrow range over Antarctica of just 10.1 to 11.3 m s$^{-1}$, 11 m s$^{-1}$ on average (Fig. 6c). Source wind10 of EAIS precipitation (10.8 m s$^{-1}$) is only marginally lower than that of WAIS (11.2 m s$^{-1}$) and AP (10.9 m s$^{-1}$), and the difference is also small for regions above and below 2250 m (10.7 vs. 11 m s$^{-1}$). This narrow range might be reflective of the role that cyclones and storm tracks play in influencing moisture availability for Antarctica

through evaporation and moisture transport (Sinclair and Dacre, 2019). Investigation of the relationship between cyclones, and other forms of storms, and this unexpectedly narrow band of source wind10 of annual mean precipitation over Antarctica is merited; however, this is outwith the scope of the present study.

     Relationships between source properties and source locations depend on several factors. Firstly, evaporation is directly dependent on wind10, rh2m, and SST (Eq. 1). Thus, whilst moisture transport paths partly control the spatial and temporal

distribution of source properties and locations, evaporation processes will cause some decoupling of moisture source properties from source locations. The clearest example of this is the impact of wind10 variability, which might relate to storm activities. Given evaporation will preferentially occur during higher wind speeds at any oceanic grid cell, moisture source wind10, which is weighted by evaporation fluxes, is larger than mean wind10 at this grid cell. Indeed, differences between source wind10 and wind10 at source are generally positive, with an Antarctic average value of 2.1 m s$^{-1}$ (Fig. 7). Whilst there are seasonal

variations in this impact of source storminess on Antarctic precipitation (Antarctic mean values are +2.9 m s$^{-1}$ in DJF, +1.6 m s$^{-1}$ in MAM, +1.3 m s$^{-1}$ in JJA, and +2.2 m s$^{-1}$ in September-October-November (SON)), the consistent 1-3 m s$^{-1}$ offset in all seasons suggest that, although the magnitude of these discrepancies must be affected by shifts in precipitation source regions, our isolation of a source storminess impact on Antarctic precipitation is robust. This is a clear example of how our new water tracer methods can be used to isolate a thermodynamic control of Southern Ocean surface wind on moisture availability

for Antarctic precipitation.

     While annual cycles of source latitude are strongly influenced by meridional thermal gradients and sea ice variations, annual cycles of source properties are additionally influenced by their seasonal variations at mid-latitudes. For example, MAM precipitation is from more southern regions than JJA precipitation likely because of sea ice retreat (Fig. B6a3); source SST of MAM precipitation is higher than that of JJA precipitation due to higher SST at mid-latitudes in MAM (Fig. B6d3, similar for

source rh2m in Fig. B6e3); and DJF precipitation comes from less windy regions than JJA precipitation - partly due to weaker westerlies in austral summer (Fig. B6f2 and B6f4).

### 3.4   How do source properties vary with precipitation rates?

Alongside geographical and seasonal variations, we examine now moisture source anomalies of heavy and light precipitation at two Antarctic sites and across Antarctica. We choose EDC and Halley as inland and coastal sites, respectively (Fig. B1b).

For the two sites, after applying a standard threshold (see Section 2.3), daily precipitation rates at each site are divided into 100 percentiles. For each percentile, the precipitation rate and its contribution to the total precipitation amount can be estimated (Fig. B7). The higher percentiles, with their larger precipitation rates, contribute a large proportion of the total site precipitation.





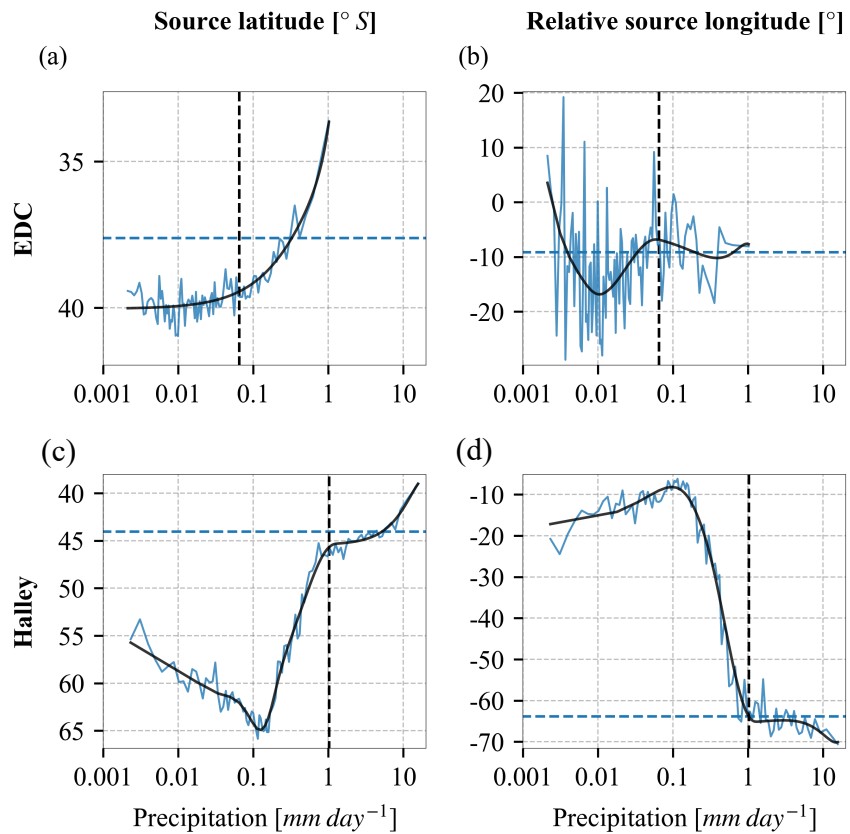

**Figure 8.** Variations of precipitation source properties with precipitation rates at (a-b) EDC and (c-d) Halley. Source properties include mass-weighted mean open-oceanic evaporative (a, c) source latitude and (b, d) relative source longitude. Precipitation rates are calculated for each percentile of daily precipitation rates. Horizontal dashed blue lines show annual mean source properties and vertical dashed black lines show annual mean precipitation rates. Solid black lines show spline fits to solid blue lines.

As a result, sources of a few top percentiles exert a strong control on the mass-weighted average source properties of total precipitation (Fig. 8).

HP over Antarctica depends mainly on intrusions of moist and warm maritime air masses. As underlying SST decreases during poleward moisture transport, surface evaporation might be suppressed. Consequently, HP would derive its moisture from more remote regions than the rest of precipitation (Terpstra et al., 2021). This hypothesis, based on a case study, is supported by our modelling results on a climatological scale. Source-sink distance anomalies of HP relative to the rest of precipitation are ∼300 km over Antarctica. By sub-regions, the source-sink anomalies are 290 km over EAIS, 330 km over

WAIS, and 670 km over AP (Fig. 9d). Source latitude anomalies of HP are 2.9° over Antarctica, 2.9° over EAIS, 3.1° over WAIS, and 4.9° over AP (Fig. 9a). These results quantify the degree to which HP is related to more distant (300 km) and equatorward (2.9°) source regions.





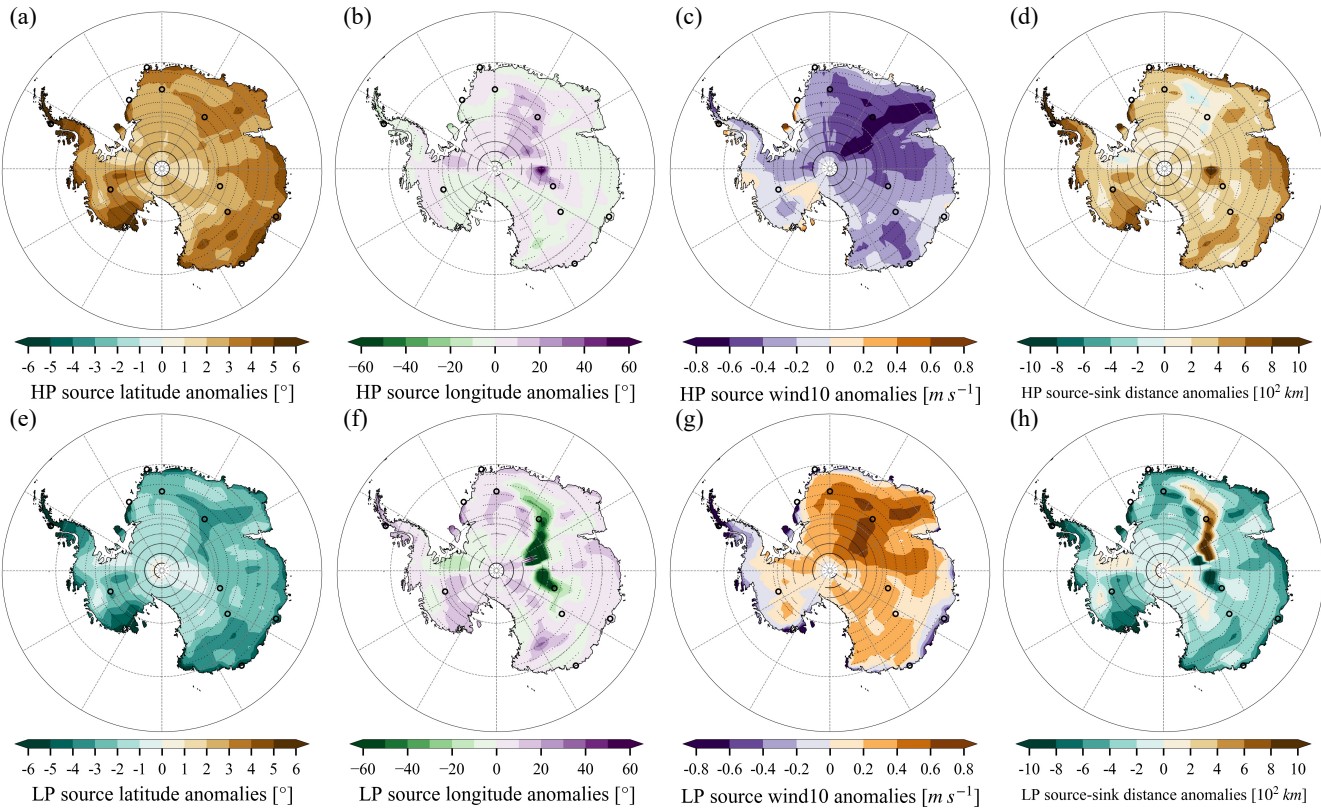

**Figure 9.** Moisture source anomalies of (a-d) HP and (e-h) LP. HP and LP source anomalies are relative to non-HP and non-LP days, respectively. Source properties include mass-weighted mean open-oceanic evaporative (a, e) source latitude, (b, f) source longitude, (c, g) source wind10, and (d, h) source-sink distance. Stippling points represent significant differences at 5% significance level based on statistical tests: for all variables except source longitude, student's t-test with Benjamini-Hochberg Procedure controlling false discovery rates (Benjamini and Hochberg, 1995) is adopted; for source longitude, Watson-Williams F-test for circular statistics (Watson and Williams, 1956) is employed. Positive source latitude difference means more equatorward, and positive source longitude difference means more eastward.





Similar features can be observed at the EDC and the Halley sites. At EDC, source latitude moves equatorward with increasing precipitation rates (from 40° S for LP to 36° S for HP), though relative source longitude indicates large fluctuations (Fig. 8). In contrast, Halley experiences two distinct precipitation regimes. For daily precipitation below $\sim$0.1 mm day$^{-1}$, moisture is derived from more poleward oceans (60° S) and undergoes less eastward transport (by 15°) than the rest of precipitation, which indicates local sources. Above $\sim$1 mm day$^{-1}$, precipitation originates from more equatorward oceans (45° S) and undergoes more eastward transport (by 65°) than the rest of precipitation, which represents remote sources. See also the histograms of source properties for a different type of depiction of this behaviour (Fig. B8).

HP also shows notable source longitude anomalies (Fig. 9b). In particular, the degree of eastward moisture advection decreases towards the Antarctic interior, reaching a $\sim$15° anomaly at Dome F. This is reflective of more direct atmospheric meridional flows during heavy precipitation events. In coastal regions, negative source longitude anomalies generally indicate remote moisture sources and thus larger zonal moisture transport by westerlies.

Furthermore, source wind10 of HP is typically smaller than that of the rest of precipitation (-0.32 m s$^{-1}$ over Antarctica, -0.36 m s$^{-1}$ over EAIS, -0.12 m s$^{-1}$ over WAIS, and -0.21 m s$^{-1}$ over AP). This is likely due to HP deriving its moisture from more equatorward oceans where wind10 is generally smaller (Fig. B2), rather than that less windy conditions favour HP.

Source property anomalies of LP generally show opposite patterns to HP: LP derives moisture from more poleward regions (-2.4° over Antarctica, Fig. 9e); source longitude shows diverse regional patterns (Fig. 9f); LP originates from more windy oceans over large parts of Antarctica (the differences average to 0.22 m s$^{-1}$ over Antarctica, Fig. 9g); and LP relies more on short-range moisture transport (the differences average to -290 km over Antarctica, Fig. 9h).

### 3.5 How does SAM affect precipitation source regions?

SAM is primarily characterised by zonal winds and is thus linked to the likelihood of meridional (versus more zonal) atmospheric moisture transport. During positive SAM phases, stronger westerlies may be associated with more local storms and evaporation; whereas negative SAM favours poleward intrusions of maritime air masses from more distant sources, due to amplified Rossby waves (Stenni et al., 2010; Schlosser et al., 2016). We thus explore impacts of SAM states on Antarctic precipitation source regions and properties.

We find that negative SAM polarity is linked with more equatorward sourced moisture over most of Antarctica (Fig. 10a). The difference in source latitude between SAM+ and SAM- months is $\sim$-2.4° over Antarctica (-2.2° over EAIS, -3.1° over WAIS, and -1.2° over AP). Effects of SAM polarity can also be observed in zonal mean source latitude of atmospheric humidity (Fig. B9). Above Antarctica, atmospheric humidity during SAM- months comes generally from more equatorward regions than SAM+ months, by up to 6°. These results quantify the degree to which poleward moisture fluxes are associated with SAM.

Impacts of SAM on source longitude vary considerably across Antarctica (-91° to 67°, Fig. 10b). Over large parts of Antarctica, SAM+ is linked with more eastward moisture transport by westerlies (source longitude differences: -17°, area-weighted over negative anomaly regions). In a few regions, *e.g.* near Vostok, SAM+ is connected to positive source longitude anomalies (8°, area-weighted over positive anomaly regions).



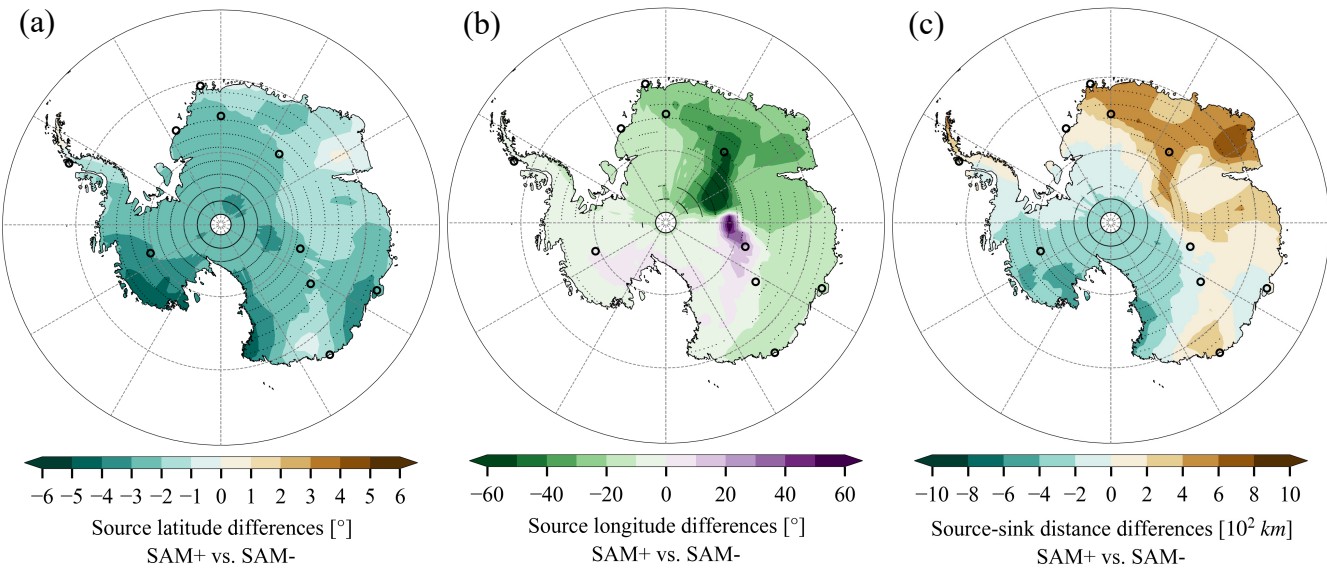

**Figure 10.** Differences in mass-weighted mean open-oceanic evaporative (a) source latitude, (b) source longitude, and (c) source-sink distance of precipitation between SAM+ and SAM- months. Monthly mean source latitude, relative source longitude, and source-sink distance are deducted from monthly values before analysis. Stippling points represent significant differences at 5% significance level based on statistical tests: for source latitude and source-sink distance, student's t-test with Benjamini-Hochberg Procedure controlling false discovery rates (Benjamini and Hochberg, 1995) is adopted; for source longitude, Watson-Williams F-test for circular statistics (Watson and Williams, 1956) is employed. Positive source latitude difference means more equatorward, and positive source longitude difference means more eastward.

Correspondingly, differences in source-sink distance between SAM+ and SAM- months exhibit a dipole pattern (Fig. 10c, -600 to 800 km). Over WAIS and southern EAIS, source latitude anomalies dominate and thus SAM+ is connected to shorter source-sink distance (-230 km, area-weighted over negative anomalies). Over northern EAIS, source longitude anomalies dominate and thus SAM+ is linked with longer source-sink distance (280 km, area-weighted over positive anomalies). This
indicates that whilst SAM states exert controls over meridional moisture fluxes, the picture is not homogenous across the whole of Antarctica (Schlosser et al., 2010, 2016).

We note that SAM- months are associated with more equatorward moisture sources than SAM+ months, and HP derives its moisture from more northern oceans than the rest of precipitation. So, does SAM exert control over the frequency and intensity of HP? In our PI simulation, there is no significant correlation between SAM and the intensity of HP across Antarctica, but SAM
does impact the frequency of HP over parts of Antarctica (Fig. B10). Correlation patterns between SAM and HP frequency are similar to those between SAM and monthly precipitation, which means SAM can influence the Antarctic precipitation amount through its controls on HP frequency.



## 4 Conclusions and perspectives

Antarctic precipitation plays a crucial role in determining global sea level. However, our understanding of its thermodynamic
and dynamic drivers is limited. Here we have begun to tackle some of the limits on our understanding through the development
and application of new, prescribed-region and scaled-flux, water tracing diagnostics in the atmospheric GCM ECHAM6. In
addition to using these diagnostics together in a novel way, we develop the dynamic (sea ice source) tracking alongside the
sub-grid-scale partitioning of fluxes. Together, these developments yield a powerful tool from which we can infer evaporative
source regions and properties of Antarctic precipitation, including moisture source locations, SST, rh2m, and wind10.

In our preindustrial ECHAM6 simulation, the contribution to Antarctic precipitation from the open ocean is determined to be
89%, and 6% from sea ice. The open ocean south of $50°$ S contributes 28%; the Atlantic Ocean north of $50°$ S contributes 10%;
the Pacific Ocean north of $50°$ S contributes 28%; and the Indian Ocean north of $50°$ S contributes 23%. Remaining contribu-
tions come from AIS (0.6%) and other continents (4.4%). While the annual cycles of these contributions are primarily driven by
variations in meridional thermal gradients and sea ice, spatial patterns are heavily influenced by atmospheric moist isentropes.
The tendency for poleward vapour transport to follow moist isentropes means that moisture from more equatorward regions
is transported at higher altitudes to more central Antarctic regions, and also that Antarctic regions at higher elevations receive
a larger proportion of precipitation from more equatorward regions compared to lower elevation areas (Bailey et al., 2019).
The mass-weighted mean open-oceanic evaporative source latitude of total precipitation averages to $\sim41°$ S over Antarctica.
Precipitation at elevations above $2250$ m originates from more equatorward ($4°$) oceans than that at elevations below $2250$ m
($38°$ S vs. $42°$ S), and EAIS precipitation is from more northern oceans by $3°$ than WAIS and AP ($40°$ S vs. $43°$ S).

Our simulated source SST of annual mean precipitation ranges from 9.8 to $16.3°$C across Antarctica, which is within the
range of existing literature estimates (Petit et al., 1991; Koster et al., 1992; Delaygue et al., 2000). Whilst our results are from
just one simulation using just one model, our methods yield a more precise value, compared to previous methods. Source
rh2m ranges from 75.6% to 83.3%, and source wind10 varies between 10.1 and $11.3$ m s$^{-1}$, in precipitation across Antarctica.
Source properties of Antarctic precipitation are highly related to source latitude, partly because meridional gradients of SST,
rh2m, and wind10 are larger than zonal gradients at mid-latitudes. Where these properties tend to decouple from each other,
this can indicate storm or seasonal controls on Antarctic precipitation sources.

Of the source properties we examine, wind10 appears to play a particularly important role in controlling Antarctic precip-
itation. The narrow range of annual mean source wind10 ($10.1$-$11.3$ m s$^{-1}$) is noteworthy, and it is consistently higher than
annual mean wind10 at precipitation source locations (by an Antarctic average value of $2.1$ m s$^{-1}$). This is likely due to higher
source wind speeds driving more evaporation and thus moisture availability, alongside possible controls on moisture transport
pathways. Since the wind field is linked to cyclone activities and large-scale circulation patterns including subtropical gyres,
further investigation is necessary to clarify these connections.

Moisture sources are related to precipitation rates and SAM. HP obtains its moisture from more equatorward sources, with
an Antarctic average shift in source regions of $2.9°$ further north and $300$ km farther away compared to the rest of precipitation.
This is consistent with the case study-based hypothesis of Terpstra et al. (2021). As speculated by Stenni et al. (2010) and



Buizert et al. (2018), negative SAM polarity is connected to more equatorward moisture provenance than positive SAM phases by an average of 2.4°. These findings might explain why SAM influences HP frequency, and thus precipitation. Like wind source effects, given that SAM can affect both moisture availability and moisture transport, further detailed water tracer-based assessment of Antarctic precipitation changes under SAM variations is also merited.

We have identified several potential directions for future research for water tracer-based studies. In addition to SAM, other large-scale atmospheric circulation indices such as zonal wave three (Raphael, 2007; Uotila et al., 2013) could be studied. Case studies combined with observations, such as water isotopes and extreme events, might also provide new insights.

While our work focuses on Antarctica in preindustrial conditions, further research will explore changes in moisture sources under different climate conditions in various regions. We note that the results presented here are based solely on a single model. To enable us to explore the model dependence of our results, we are developing similar water tracing diagnostics in another atmospheric GCM, the UK Met Office Unified Model (Brown et al., 2012). Comparing water tracing results between different models will enable quantification of the model dependency of these results. Alongside climate and model dependency, impacts of model resolution on the results also merit further study. Finally, we note these new scaled-flux tracing approaches are not only applicable to water tracers in atmospheric GCMs but could also be applied to other types of tracers in a numerical system: the full potential of our water tracing diagnostics is yet to be identified.

*Code and data availability.* The ERA5 reanalysis can be obtained from the Climate Data Store (https://cds.climate.copernicus.eu). The Antarctic accumulation reconstruction from Medley and Thomas (2018) is available here: https://earth.gsfc.nasa.gov. The SAM index compiled by Marshall (2003) is available here: https://legacy.bas.ac.uk/met/gjma/sam.html. The Bedmap2 product created by Fretwell et al. (2013) is available here: https://www.bas.ac.uk/project/bedmap-2. The division of Antarctica is available here: http://imbie.org/imbie-2016/drainage-basins. The AMIP SST and SIC dataset is available here: https://esgf-node.llnl.gov/search/input4mips. The ECHAM6 simulation output and data analysis scripts are available from the authors upon reasonable request.



**Appendix A: Implementation of scaled-flux tracers in ECHAM6 and comparison with prescribed-region tracers**

Here we introduce our scaled-flux water tracing approach and then compare its results against the prescribed-regions water
tracing method. The basic idea of this method follows Fiorella et al. (2021, see their section 2.1), but our implementation is
designed to ensure that the tracing water budget is closed.

In our scaled-flux water tracing approach, three water tracers ($wt1$, $wt2$, $wt3$) are required for each evaporative source
condition (*e.g.* source latitude). The combination of $wt1$ and $wt2$ track the amount of water sourced from the open ocean,
while $wt3$ follows water evaporated from both land and sea ice. All the water in the model is therefore tracked by the sum of
these three tracers.

Upward evaporative fluxes of tracer water are scaled based on evaporation conditions. For any evaporative flux from the
open ocean, $E_i^{ocn}$, the corresponding tracer evaporative flux of $wt1$ is calculated as

$$E_i^{wt}(wt1, t_i, \lambda_i, \phi_i) = E_i^{ocn}(t_i, \lambda_i, \phi_i) \times SF(wt1, t_i, \lambda_i, \phi_i), \tag{A1}$$

where $t$ denotes time, $\lambda$ longitude, and $\phi$ latitude. The scaling factor $SF(wt, t, \lambda, \phi)$ is defined for $wt1$ as

$$SF(wt1, t, \lambda, \phi) = \begin{cases} \frac{X(t,\lambda,\phi) - X_{lower}}{X_{upper} - X_{lower}} & \text{over the open ocean,} \\ 0 & \text{over land and sea ice,} \end{cases} \tag{A2}$$

where $X$ is the source property of interest. $X_{lower}$ and $X_{upper}$ are two constants set to a lower and upper limit of $X$ to
ensure $SF$ remains in the range of (0, 1). As arithmetic operations cannot be applied to circular data directly, tracers for
source longitude are scaled based on the sine and cosine of longitude. Thereafter, source longitude is estimated according to
trigonometrical functions. Values of $X_{lower}$ and $X_{upper}$ are defined as [-90°, 90°] for latitude, [-1, 1] for sine and cosine of
longitude, [-5°C, 45°C] for SST, [0, 160%] for rh2m, and [0, 28 m s$^{-1}$] for wind10.

The second water tracer ($wt2$) is defined such that the sum of $wt1$ and $wt2$ tracks the total open ocean evaporation. Therefore,
the evaporative flux for $wt2$ is given by Eq. (A1) but with the scaling factor $SF(wt2, t, \lambda, \phi)$ set as

$$SF(wt2, t, \lambda, \phi) = \begin{cases} 1 - SF(wt1, t, \lambda, \phi) & \text{over the open ocean,} \\ 0 & \text{over land and sea ice,} \end{cases}$$

which gives, $E_i^{wt}(wt1, t_i, \lambda_i, \phi_i) + E_i^{wt}(wt2, t_i, \lambda_i, \phi_i) = E_i^{ocn}(t_i, \lambda_i, \phi_i)$. Note, downward condensation fluxes of tracer water
at the surface are proportional to normal water fluxes as in the predefined-region water tracing approach.

For the atmospheric specific humidity $q_i^{ocn}(t, p, \lambda, \phi)$ formed from the evaporation flux $E_i^{ocn}(t_i, \lambda_i, \phi_i)$, we have the corre-
sponding water tracer quantity,

$$q_i^{wt}(wt1, t, p, \lambda, \phi) = q_i^{ocn}(t, p, \lambda, \phi) \times SF(wt1, t_i, \lambda_i, \phi_i), \tag{A3}$$

where $p$ is the pressure level. By summing up all vapour contributions in a grid box, we obtain

$$\sum_i q_i^{wt}(wt1, t, p, \lambda, \phi) = \sum_i \left( q_i^{ocn}(t, p, \lambda, \phi) \times SF(wt1, t_i, \lambda_i, \phi_i) \right). \tag{A4}$$



where $\sum_i q_i^{wt}(wt1, t, p, \lambda, \phi)$ is the atmospheric water tracked by $wt1$.

By substituting $SF(wt1, t_i, \lambda_i, \phi_i)$ from Eq. (A2) into equation Eq. (A4) and rearranging, we can obtain the following expression for the mass-weighted mean open-oceanic evaporative source property of the atmospheric water,

$$\frac{\sum_i \left(q_i^{ocn}(t,p,\lambda,\phi) \times X(t_i,\lambda_i,\phi_i)\right)}{\sum_i q_i^{ocn}(t,p,\lambda,\phi)} = \frac{\sum_i q_i^{wt}(wt1,t,p,\lambda,\phi)}{\sum_i q_i^{ocn}(t,p,\lambda,\phi)} \times (X_{upper} - X_{lower}) + X_{lower}. \tag{A5}$$

In the above equation, $\sum_i q_i^{ocn}(t,p,\lambda,\phi)$ is the atmospheric water sourced from the open ocean and can be replaced with the sum of $wt1$ and $wt2$, which gives

$$\frac{\sum_i \left(q_i^{ocn}(t,p,\lambda,\phi) \times X(t_i,\lambda_i,\phi_i)\right)}{\sum_i q_i^{ocn}(t,p,\lambda,\phi)} = \frac{\sum_i q_i^{wt}(wt1,t,p,\lambda,\phi)}{\sum_i q_i^{wt}(wt1,t,p,\lambda,\phi) + \sum_i q_i^{wt}(wt2,t,p,\lambda,\phi)} \times (X_{upper} - X_{lower}) + X_{lower}. \tag{A6}$$

As passive water tracers always follow normal water proportionally after evaporation, evaporative source properties of precipitation can be obtained in the same way.

The third water tracer ($wt3$) is used to track the water evaporated from land and sea ice, hence, $SF(wt3, t, \lambda, \phi) = 0$ over the open ocean, and $SF(wt3, t, \lambda, \phi) = 1$ over land and sea ice. Therefore, the combination of the three tracers tracks all the water in the model. This allows a correction to be applied at each grid point and timestep to ensure that the sum of the three water tracers does not deviate from normal water in the model. Small deviations occur for numerical reasons related to partitioning normal water into multiple water tracers and they can accumulate and propagate. We applied corrections to atmospheric tracer water to ensure their sum equals normal water. Importantly, the proportion of each water tracer does not change after corrections. These corrections are applied to both water tracing methods. The magnitude of corrections is at an acceptable level (less than 2‰).

The atmospheric tracer water content is initialised as a product of atmospheric normal water content and the scaling factor $SF(wt, 0, \lambda, \phi)$.

To evaluate the precision of the scaled-flux water tracing method, we compare it against results from the predefined-region water tracing method. The latter can also be used to estimate evaporation source properties (Koster et al., 1992; Delaygue et al., 2000). Briefly, we divide the global open ocean into multiple tagging regions based on values of source properties (*e.g.* latitude bins from -90° to 90°, every 10°) at each time step and trace the water evaporated from these individual regions. We then estimate mass-weighted mean source properties from all these prescribed-region water tracers. Note that we have to approximate the source property of each water tracer as the middle value of the variable bin (*e.g.* 5° for the latitude bin 0-10°).

The two approaches deliver quite similar results. For example, Fig. B11 compares source latitude of annual mean precipitation inferred from the two methods. The maximum absolute difference between the two methods is less than 2.8°, and the mean absolute difference is 0.6°. For the prescribed-region tracers, there are biases due to the approximation of values of source properties as the middle value of each bin. These biases are visible as colour strips in Fig. B11c and can be reduced by decreasing sizes of variable bins (*e.g.* using 5° latitude bins). However, that would require even more computational resources for the prescribed-region water tracers. The scaled-flux water tracing method needs only three tracers for each source property, whereas the prescribed-region water tracing approach demands 18 (36) tracers for source latitude with 10° (5°) latitude bins.



Note that each water tracer requires ~10% additional computational time. Equivalent comparisons of source longitude, SST, rh2m, and wind10 from the two approaches reveal similar patterns. These results highlight the benefits of using the efficient
425   scaled-flux tracer method.



## Appendix B: Additional figures

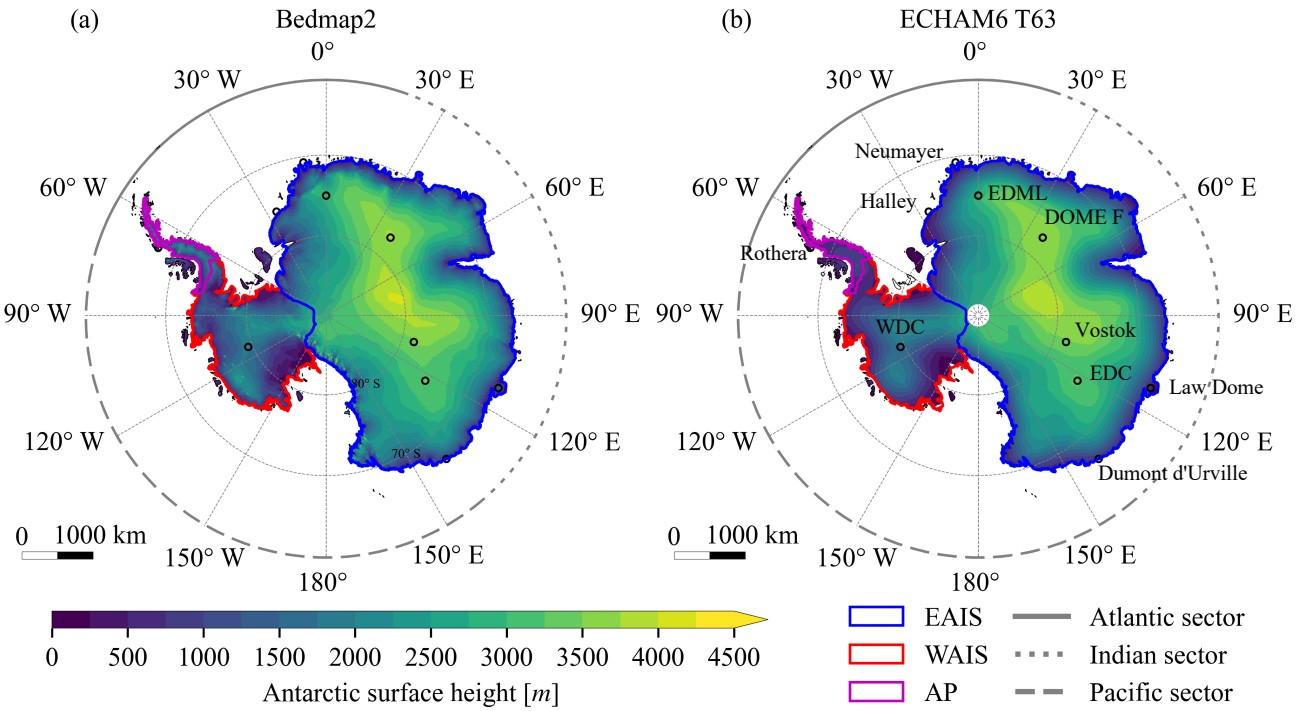

**Figure B1.** Antarctic surface height (a) in the observation-based Bedmap2 product (Fretwell et al., 2013) and (b) in our model simulation with T63 resolution. Definitions of EAIS, WAIS, and AP are based on the work of E. Rignot and J. Mouginot (http://imbie.org/, last access date: 20 Feb 2023). Oceanic sectors are specified as below: Atlantic sector (70° W to 20° E), Indian sector (20° E to 140° E), and Pacific sector (140° E to 70° W). Locations of five inland and five coastal sites are indicated with black empty circles. EDC stands for EPICA Dome Concordia, EDML for EPICA Dronning Maud Land, Dome F for Dome Fuji, and WDC for the WAIS Divide ice core.





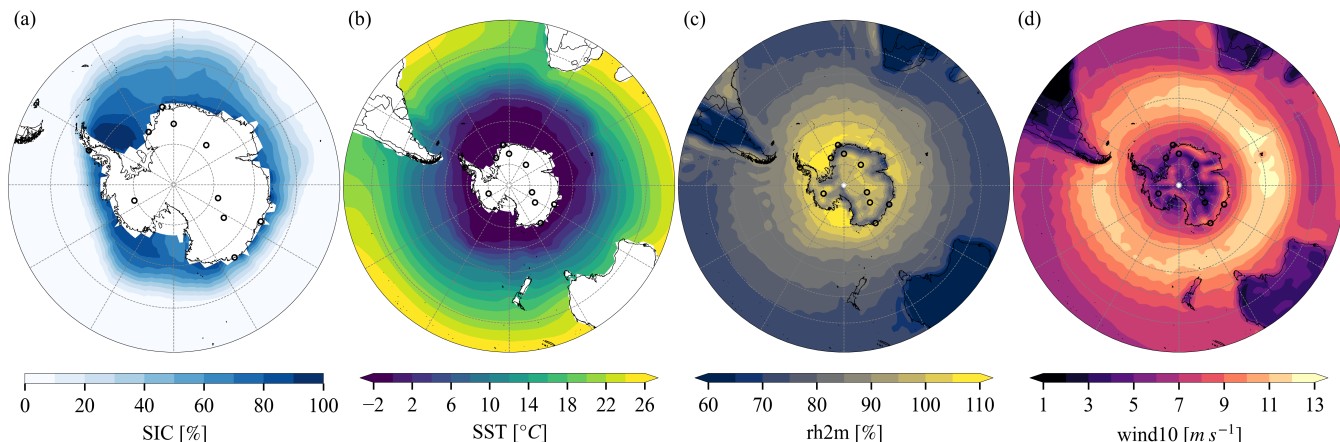

**Figure B2.** Annual mean (a) sea ice concentration (SIC), (b) sea surface temperature (SST), (c) 2-meter relative humidity (rh2m), and (d) 10-meter wind speed (wind10) in the simulation.



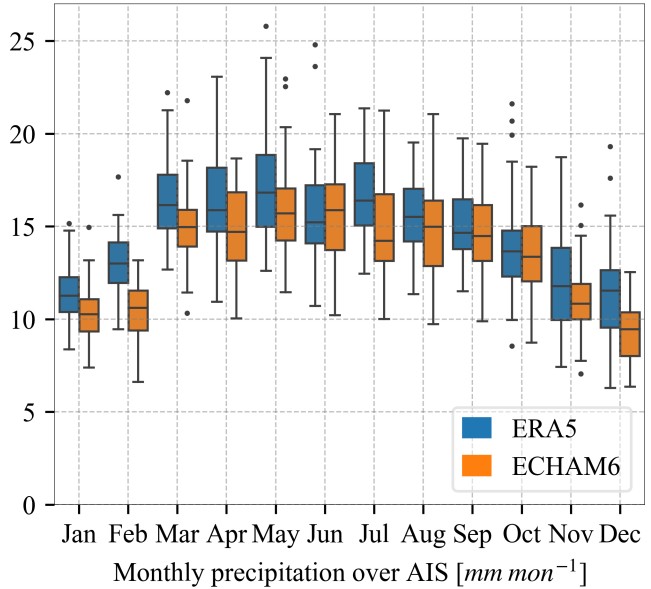

**Figure B3.** Monthly mean precipitation over Antarctica in ERA5 (1979-2021) and the ECHAM6 preindustrial simulation.



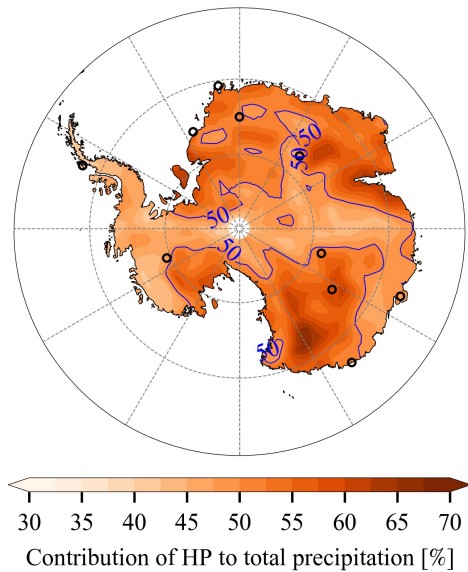

**Figure B4.** Contribution of HP to total precipitation in our ECHAM6 preindustrial simulation. Blue lines show 50% contours of the contribution.



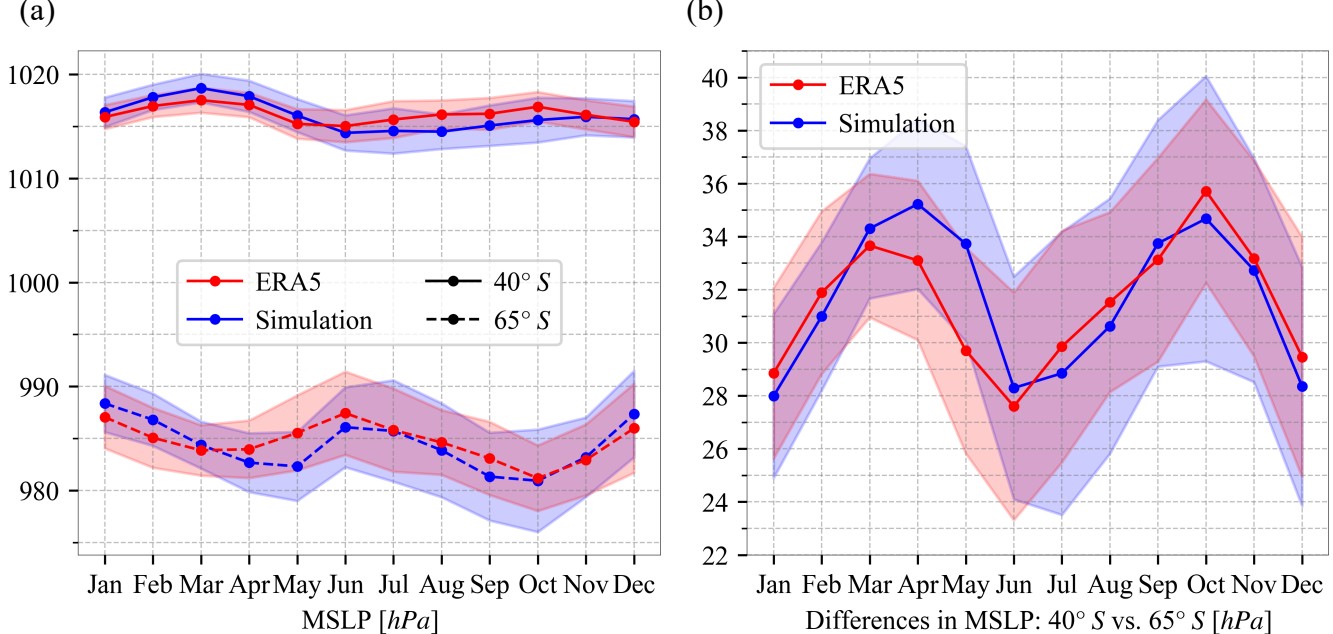

**Figure B5.** (a) Monthly mean zonal mean sea level pressure (MSLP) at 40° S and 65° S in ERA5 (1979-2021) and our simulation. (b) Differences in monthly mean zonal mean sea level pressure between 40° S and 65° S in two datasets. The colour shadings show one standard deviation. Simulation results deviate less than one standard deviation from ERA5. Root mean squared errors between simulated and assimilated MSLP at 40° S and 65° S, and their differences, are 1.0, 1.4, and 1.5 $hPa$, respectively.



**Figure B6.** Mass-weighted mean open-oceanic evaporative (a) source latitude, (b) relative source longitude, (c) source-sink distance, (d) source SST, (e) source rh2m, and (f) source wind10 of the annual mean (the 1st column) and seasonal mean (the 2nd to 5th columns) precipitation. Red lines show contours of one standard deviation. DJF refers to December-January-February, MAM March-April-May, JJA June-July-August, and SON September-October-November.



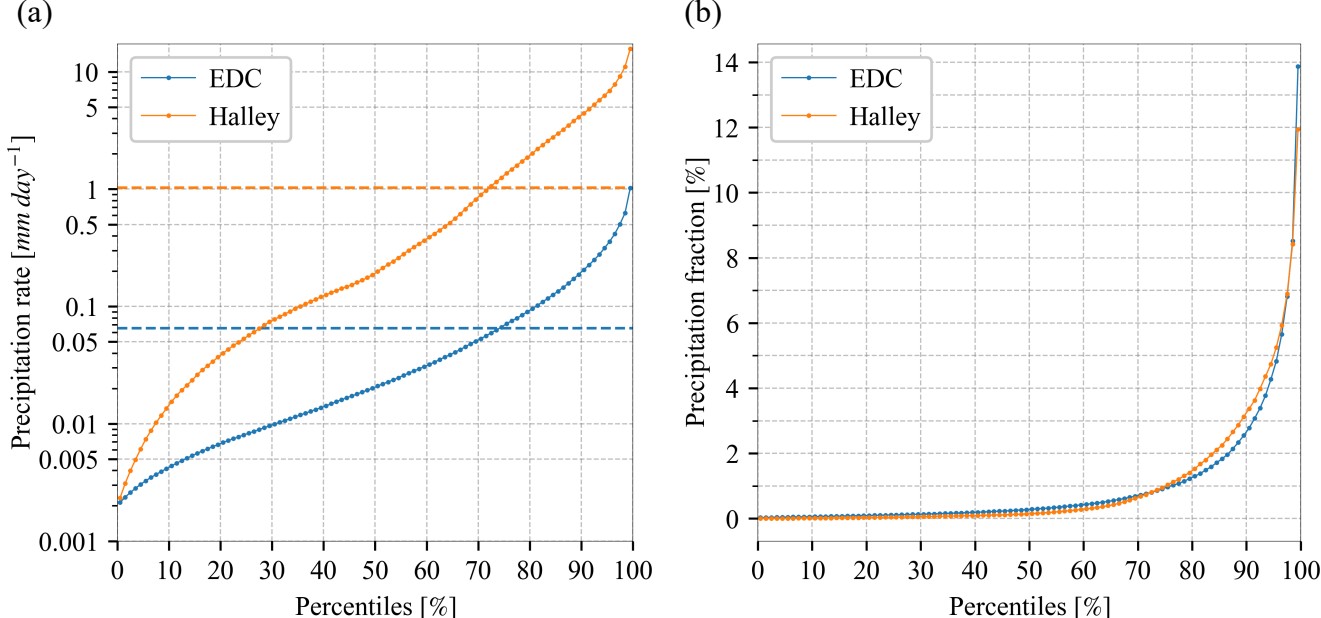

**Figure B7.** (a) Precipitation rates and (b) contributions to total precipitation of each percentile (from 0% to 100%, every 1%) of daily precipitation rates at EDC and Halley. Horizontal coloured dash lines in (a) indicate annual mean precipitation rates.



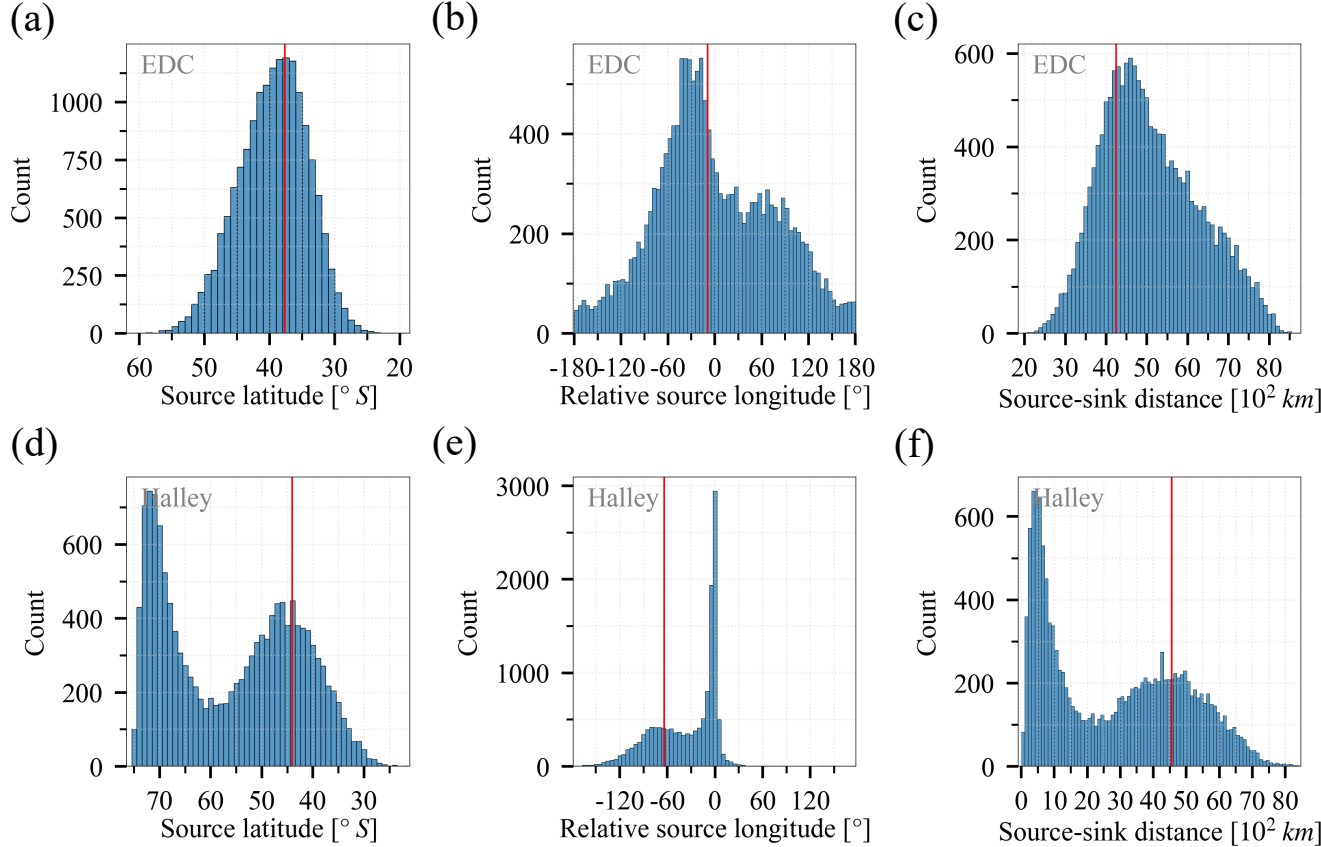

**Figure B8.** Histograms of source properties of daily precipitation at (a-c) EDC and (d-f) Halley. Source properties include mass-weighted mean open-oceanic evaporative (a, d) source latitude, (b, e) relative source longitude, and (c, f) source-sink distance. Vertical red lines represent source properties of annual mean precipitation.




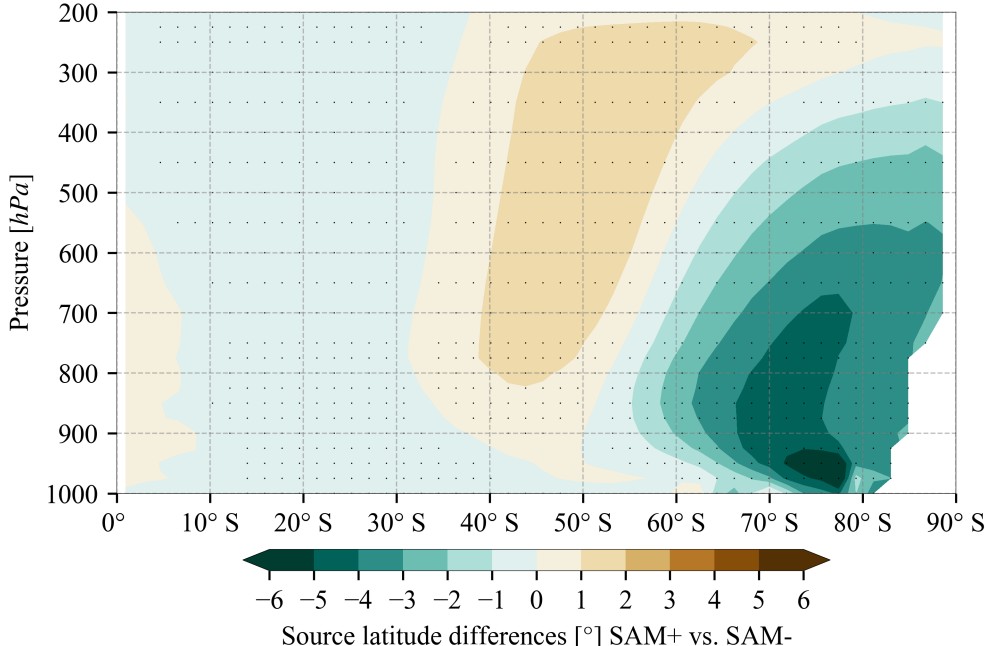

**Figure B9.** Differences in zonal-averaged source latitude of atmospheric humidity between SAM+ and SAM- months. Monthly mean source latitudes are deducted from monthly values before analysis. Stippling points represent significant differences at 5% significance level based on the student's t-test with Benjamini-Hochberg Procedure controlling false discovery rates (Benjamini and Hochberg, 1995). Positive source latitude difference means more equatorward.



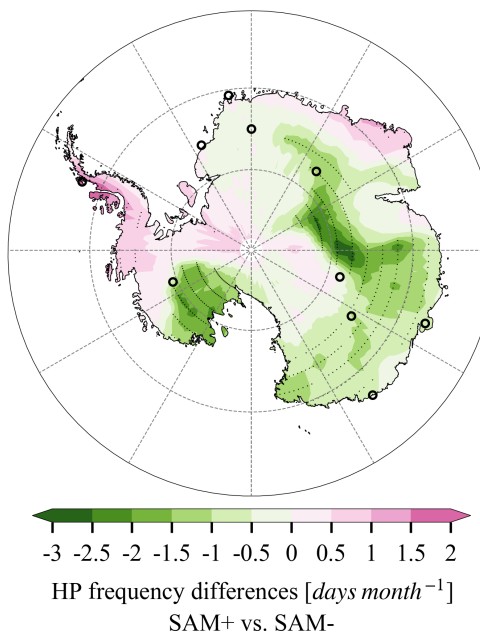

**Figure B10.** Differences in HP frequency between SAM+ and SAM- months. Stippling points represent significant differences at 5% significance level based on the student's t-test with Benjamini-Hochberg Procedure controlling false discovery rates (Benjamini and Hochberg, 1995).



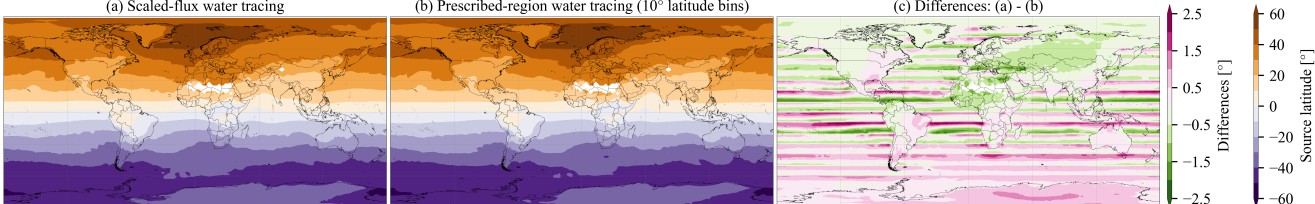

**Figure B11.** Mass-weighted mean open-oceanic evaporative source latitude of annual mean precipitation estimated from (a) the scaled-flux water tracing approach and (b) the prescribed-region water tracing approach using 10° latitude bins. (c) The differences between (a) and (b). We utilised only one-year simulation data here with a one-year spin-up period to save computational resources. Positive source latitude difference means more equatorward.



*Author contributions.* QG, LCS, MW, and AM together co-led the development of this study. QG, MW, and XS implemented the water tracers in ECHAM6, with guidance from LCS and AM. QG ran the simulation, performed all data analysis and wrote with LCS the first draft of this manuscript. All authors contributed to the final draft.

430 *Competing interests.* The authors declare that they have no conflict of interest.

*Acknowledgements.* This publication was generated in the frame of DEEPICE project. The project has received funding from the European Union's Horizon 2020 research and innovation programme under the Marie Sklodowska-Curie grant agreement No 955750. QG and MW acknowledge the technical support and computing resources provided by the AWI Computer and Data Center in setting up and running the ECHAM6 simulations. AM and LCS were supported by: the European Union's Horizon 2020 research and innovation programme

435 under Grant agreement no. 820970 (TiPES project; this paper is TiPES contribution number 224) and LCS also from the NERC National Capability International grant SURface FluxEs In AnTarctica (SURFEIT): NE/X009319/1; NE/X009386/1; and NE/P009271/1 grants. EC acknowledges the financial support from the French National Research Agency under the "Programme d'Investissements d'Avenir" (ANR-19-MPGA-0001) through the Make Our Planet Great Again HOTCLIM project. XS is supported by the National Natural Science Foundation of China (NSFC) (grant no. 42206256).



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
