# Peer review of "Evaporative controls on Antarctic precipitation: An ECHAM6 model study using novel water tracer diagnostics"

_EGUsphere, 2023_

## Referee Comment (RC1)

**Review of "Evaporative controls on Antarctic precipitation: An ECHAM6 model study using novel water tracer diagnostics" by Gao et al., 2023 submitted to The Cryosphere**

This study quantifies the evaporative sources of Antarctic precipitation for a preindustrial simulation with the ECHAM6 model. The spatial and seasonal variability of the evaporative moisture source contributions to Antarctic precipitation at different elevations and for precipitation events of different intensities are discussed along with the typical moisture transport pathways to the ice sheet. Different moisture source conditions relevant for setting the evaporation flux and impacting water isotope records as well as the anomalies of these source conditions during uptakes contributing to Antarctic precipitation compared to climatology are investigated. Finally, shifts in moisture source regions associated with variations in the Southern Annular mode are analysed. The chosen methodological approach is based on an innovative scaled-flux water tracer approach implemented in ECHAM6 similarly as initially proposed by Fiorella et al. 2021 using the iCAM6 global circulation model. I much enjoyed reading this interesting and innovative paper, which is well-written and has nice and captivating figures. For me personally the scientific highlights of this paper are i) the anomalously strong storminess at the moisture source of humidity feeding Antarctic precipitation and ii) the shifts in source locations and conditions observed with the SAM. On a methodological side, I find the scaled flux water tracer implementation very attractive and the documentation of their implementation in ECHAM6 in Appendix A useful. I particularly like the comparison done with the traditional numerical tracer setup using pre-specified evaporative regions (Appendix A).

I have only one "easy" general comment, which is related to the fact that this study is based on preindustrial simulations. This is all fine per se, but the authors should make this much clearer in their introduction, in which anticipated future changes with global warming are mainly discussed. Currently, for me as a reader there is a mismatch between the use of a preindustrial simulation and the knowledge gap uncovered in the introduction with sentences like "It is not yet clear how SAM variations and associated changes in moisture flux tracks will impact precipitation across Antarctica". One cannot address this question with a pre-industrial simulation, but of course the mechanisms linking variations in SAM with changes in moisture sources and transport pathways can be studied very well with a preindustrial simulation and the authors do it elegantly. Even more importantly, the fact that the authors use a simulation with preindustrial climate conditions matters, when they compare their moisture source decomposition with other studies such as the Lagrangian study by Sodemann and Stohl, 2009. I would therefore suggest to smooth out this mismatch by pointing more at the need for a better process understanding in the introduction, and remind the reader of the different time period covered in their simulations when comparing their results to previous studies.

Minor comments:
1) "novel" in the title is a bit unspecific, could be more precise
2) L. 1: "for gaining insights into **past and future** polar, and global changes"
3) L. 1: "changes" in what exactly? Environmental changes?
4) L. 7-10 and results section about Fig. 4: "The tendency of poleward vapour transport to follow moist isentropes means that central Antarctic precipitation is sourced from more equatorward (distant) sources via elevated transport pathways than coastal Antarctic precipitation. We find however this tendency breaks down in the lower

troposphere, likely due to diabatic cooling." I find this analysis based on Fig. 4 interesting but also very puzzling. Fig. 4 shows a zonally averaged mass-weighted vertical cross section of the source latitude of water vapour. This Fig. is discussed in the result section in a moist isentropic framework, from which we would expect moisture from a given latitude to follow the moist isentrope corresponding to the surface equivalent potential temperature from that latitude. I am a not so sure about what Fig. 4 tells us exactly:

To me it seems like the moist isentropic framework is a very crude approximation to the typical transport pathways and doesn't provide more than a justification for the fact that precipitation falling on the plateau tends to come from further equatorward than coastal precipitation. Similarly, it most likely explains why precipitation in the warm season comes from further equatorward than in the cold season. But other than that, when looking at Fig. 4, I see mainly deviations from the moist isentropic framework. On the Antarctic plateau the highest elevations don't pool their moisture from the most equatorward/warmest sources. And even more generally, in the upper troposphere, I see substantial deviations of the steepness of the source latitude contours from the moist isentropes. Of course, we expect that because in these dry upper tropospheric regions water vapour can be substantially older and make the distribution of source latitudes much wider. Could the authors maybe provide a weighted mean standard deviation or interquartile range of the source latitude in addition to the mean? This would provide a way of characterising the widths of the source distributions. I would assume that it is widest in the center of the storm track but maybe I am wrong.

$\rightarrow$ Thus, in short, from this analysis, I see mainly deviations from the moist isentropic poleward moisture transport framework, rather than agreement with it. I think this aspect ought to be discussed more in depth along with Fig. 4 (both panels). And, in particular, if diabatic processes are invoked for explaining these deviations, then why not name, which one the authors think could play a role?

5) L. 14: "wind10": this variable is not yet defined in the abstract, please be specific at this stage. I also think that it is not such an elegant variable name for the main text.

6) L. 63: "issues with the **Lagrangian** identification of precipitation events solely using thresholds in specific humidity changes". Yes, true, but reproducing precipitation events at the right place and at the right time is tricky as well for ECHAM6. Actually, as long as the precipitation statistics are faithfully reproduced over the ice sheet over the time period considered, this aspect of precipitation event representation is irrelevant in the context of a free running simulation. So, I am not sure this is a fair point to make at this stage. I would simply say that the Eulerian method presented in this study is complementary to the Lagrangian one and offers an elegant online diagnostic of the moisture sources of Antarctic precipitation.

7) L. 81: "revised definitions of Heavy Precipitation": revised compared to what? I would remove "revised". Maybe also think about not using any abbreviation (and not necessarily capital letters) for heavy and light precipitation because there are already many abbreviations used in this paper.

8) L. 92: "the EAIS may be slightly too low in elevations" Why may? Some "valleys" may be missing too in the topography at this coarse resolution. Do you mean that the EAIS is on average too low?

9) L. 102: is $q^{near\_sfc}$ the $q$ at the lowest model level?

10) L. 120: to characterise the near-surface humidity gradient $q_{near\_surface}$-$q_{SST}$ or $RH_{SST}$ would be more effective, see Aemisegger and Papritz et al. 2018
Aemisegger, F., and L. Papritz, 2018: A Climatology of Strong Large-Scale Ocean Evaporation Events. Part I: Identification, Global Distribution, and Associated Climate Conditions. J. Climate, 31, 7287–7312, https://doi.org/10.1175/JCLI-D-17-0591.1.

11) L. 125: "moisture parcels" -> moisture in air parcels?

12) L. 141: "The difference…" in what? In the chosen definition of LP and HP?

13) Section 2.3: Do I understand it correctly that the LP and HP definitions are chosen such that these categories contribute to a significant share of the mass balance? Just to be symmetric in the information given about the two categories: how much does the HP category contribute to total precipitation? (i.e. what is the top-10%-precipitation-days' share of total precipitation?).

14) L. 154: "on the low side" -> lower than the reconstruction, "high across some coastal areas" -> larger than in the reconstruction

15) L. 157: I would say that ECHAM6 clearly shows a larger interannual variability (2x larger than the reconstruction): is this expected given the temporal resolution of the ice core data?

16) L. 160: mention the ERA5 period with which you compared your preindustrial simulation. The warm season precipitation (NDJFMA) is quite substantially higher in ERA5, is this an effect of the slightly warmer Southern Hemisphere atmosphere in the period 1979-2021?

17) L. 169-170: make clear that this is in the annual mean.

18) L. 170: The share of moisture sourced from sublimation over Antarctica probably depends on the parametrisation of the surface sublimation flux (e.g. Gerber et al. 2023). Could it be that the regions affected by very high sublimation fluxes with e.g. strong katabatic winds and blowing snow sublimation tend to feed cold air outbreaks and contribute more to precipitation over the ocean? Or are the sublimation fluxes over Antarctica just so small compared to the available moisture in the atmosphere? Gerber, F., Sharma, V., & Lehning, M. (2023). CRYOWRF—Model evaluation and the effect of blowing snow on the Antarctic surface mass balance. Journal of Geophysical Research: Atmospheres, 128, e2022JD037744, https://doi.org/10.1029/2022JD037744.

19) L. 179: does this finding about the precipitation coming from the open ocean south of 50°S also relate to the fact that the ocean surface of this oceanic region is larger around the AP and WAIS than for the EAIS? Does it have to do with the sea ice extent in the different basins? Does the steeper topography of the EAIS play a role (forcing the rain out of the humidity sourced from the South Atlantic south of 50°S along the EAIS slopes). Also, in terms of dynamical drivers of this share of precipitation from south of 50°S: it is really interesting to note that the Dronning Maud land receives the most equatorward moisture of whole Antarctica (even though the highest elevation of the Plateau lies much further to the southeast). This might be linked to the spiral shaped form of the Southern Ocean storm track. In the South Atlantic many extratropical cyclone genesis points are climatologically located relatively far North compared to the South Pacific (see Wernli and Schwierz, 2006). Cyclones likely play a key role in poleward moisture transport in this region.

Wernli, H., and C. Schwierz, 2006: Surface Cyclones in the ERA-40 Dataset (1958–2001). Part I: Novel Identification Method and Global Climatology. J. Atmos. Sci., 63, 2486–2507, https://doi.org/10.1175/JAS3766.1.

20) L. 186 & L. 194: "tends to take an elevated path" -> "has to rise to higher altitudes"? An elevated pathway nearly sounds as if it would travel in the upper troposphere for a very long time before raining out over the Antarctic ice sheet.

21) L. 189: "so" -> "therefore"

22) L. 191: "approximating a moist adiabatic poleward ascent", this tendency is discussed already in Stohl and Sodemann, 2009, albeit with isentropes (not moist isentropes) Stohl, A., and Sodemann, H. (2010), Characteristics of atmospheric transport into the Antarctic troposphere, J. Geophys. Res., 115, D02305, doi:10.1029/2009JD012536.

23) L. 191: "this tendency to follow contours of equivalent potential temperature breaks down in the lower troposphere" -> tendency to follow moist isentropes?

24) L. 191: about the "break-down" of the moist isentropic approximation of poleward transport: Which diabatic cooling mechanism do the authors think plays a role? Could diabatic heating especially in the upper troposphere also explain part of the observed deviations from the moist isentropic framework?

25) L. 197: as mentioned in my general comment: just make sure that it stays in the mind of the reader that this study covers another time period.

26) L. 201: most equatorward sources for DJF: that is surprising because of the minimal sea ice extent and the slightly weaker jet and storm track in summer. Is this simply due to the warmer atmosphere on average, leading to higher humidity contents and longer transport distances? And as you write flatter isentropes "giving access" to more equatorward sources?

27) L. 203: milder->flatter

28) L. 209: The fact that source longitude shows the largest interannual variability is interesting and shows that the strength of the westerlies and the storm track dynamics is likely important for modulating the moisture source properties.

29) L. 211: "source latitude and longitude"

30) L. 214: "lies within the range of estimates from the literature"

31) L. 220-227: Interesting! This corresponds to the range of wind speeds associated with events of strong ocean evaporation in the Southern Ocean as discussed by Aemisegger and Papritz, 2018. Extratropical cyclones and trailing fronts were identified as key weather systems with which enhanced ocean evaporation is associated.

32) L. 226: "other forms of storms" what is meant here: tropical cyclones or polar lows? I think if you write "extratropical cyclones propagating along the Southern Ocean storm track" you also include subgroups such as mesocyclones or polar lows.

33) L. 230: what does "evaporation processes will cause some decoupling of moisture source properties" mean? Do you mean that synoptic-scale variability at the source causes variability in the evaporative conditions?

34) L. 234: what do you mean by "wind10 at source" -> climatological wind 10 at the source?

35) L. 239: in the previous two paragraphs you just discussed **dynamic** not thermodynamic controls on moisture availability for Antarctic precipitation.

36) L. 256: A case study discussing this suppression of ocean evaporation during polarward moisture transport in warm sectors of extratropical cyclones is Thurnherr and Aemisegger, 2022.

Thurnherr, I. and Aemisegger, F.: Disentangling the impact of air–sea interaction and boundary layer cloud formation on stable water isotope signals in the warm sector of a Southern Ocean cyclone, Atmos. Chem. Phys., 22, 10353–10373, https://doi.org/10.5194/acp-22-10353-2022, 2022.

37) L. 282-285: The impact of the SAM on the storm track dynamics and shifts in strong ocean evaporation patterns as well as near surface conditions is discussed in Aemisegger and Papritz, 2018.

38) L. 291: "These results quantify the degree to which poleward moisture fluxes are associated with the SAM" -> are modulated by the SAM? And maybe instead of fluxes rather use poleward moisture transport?

39) Fig. 10: the stippling is very difficult to see

40) L. 300: "SAM states exert controls" -> "the SAM impacts"

41) L. 310: "limits on" -> "of"

42) L. 328: "yields a more precise value compared to previous method" -> explain why explicitly

43) L. 330-332: this is a bit obscure to me. True in close approximation for SST but not necessarily for rh2m and the wind speed.

44) L. 355: "other types of tracers in numerical systems" -> what do you mean by this?

45) L. 356: "the full potential of our water tracing diagnostics is yet to be identified" -> This last sentence is a bit unspecific for the end of such a nice paper.

46) Appendix A: very nice and interesting documentation. I just get lost at line 385. What does the index $i$ represent and what do you mean when you write "by summing up all vapour contributions in a grid box, we obtain:"? Do you sum up over a number of grid points in a given region? In equation A4 some variables have an index $i$ and others not. I didn't get why. And in A1 $t$, $\lambda$, $\phi$ have an index $i$, but in the next lines not. Sorry to be picky, but I really would like to understand this Appendix.

47) Appendix A: I like your comparison with the predefined-region water tracing method described at lines 410ff a lot. To me this is a highly effective way to show the advantages of the scaled-flux approach and I would find it very helpful to have a short version of the results from this comparison in the main text.

---

## Author Comment (AC1)

**Responses to comments of Referee #1**

Thank you for the time and thoughts on this manuscript. We appreciate these comments.

This study quantifies the evaporative sources of Antarctic precipitation for a preindustrial simulation with the ECHAM6 model. The spatial and seasonal variability of the evaporative moisture source contributions to Antarctic precipitation at different elevations and for precipitation events of different intensities are discussed along with the typical moisture transport pathways to the ice sheet. Different moisture source conditions relevant for setting the evaporation flux and impacting water isotope records as well as the anomalies of these source conditions during uptakes contributing to Antarctic precipitation compared to climatology are investigated. Finally, shifts in moisture source regions associated with variations in the Southern Annular mode are analysed. The chosen methodological approach is based on an innovative scaled-flux water tracer approach implemented in ECHAM6 similarly as initially proposed by Fiorella et al. 2021 using the iCAM6 global circulation model. I much enjoyed reading this interesting and innovative paper, which is well-written and has nice and captivating figures. For me personally the scientific highlights of this paper are i) the anomalously strong storminess at the moisture source of humidity feeding Antarctic precipitation and ii) the shifts in source locations and conditions observed with the SAM. On a methodological side, I find the scaled flux water tracer implementation very attractive and the documentation of their implementation in ECHAM6 in Appendix A useful. I particularly like the comparison done with the traditional numerical tracer setup using pre-specified evaporative regions (Appendix A).

I have only one "easy" general comment, which is related to the fact that this study is based on preindustrial simulations. This is all fine per se, but the authors should make this much clearer in their introduction, in which anticipated future changes with global warming are mainly discussed. Currently, for me as a reader there is a mismatch between the use of a preindustrial simulation and the knowledge gap uncovered in the introduction with sentences like "It is not yet clear how SAM variations and associated changes in moisture flux tracks will impact precipitation across Antarctica". One cannot address this question with a pre-industrial simulation, but of course the mechanisms linking variations in SAM with changes in moisture sources and transport pathways can be studied very well with a preindustrial simulation and the authors do it elegantly. Even more importantly, the fact that the authors use a simulation with preindustrial climate conditions matters, when they compare their moisture source decomposition with other studies such as the Lagrangian study by Sodemann and Stohl, 2009. I would therefore suggest to smooth out this mismatch by pointing more at the need for a better process understanding in the introduction, and remind the reader of the different time period covered in their simulations when comparing their results to previous studies.

We fully agree with the referee that model simulations across different climate periods are required while applying water tracers in the context of global warming. As the referee pointed out, we focus here on the mechanisms that imprint on moisture sources. These will be further tested and evaluated in subsequent simulations. In our revised manuscript, we stressed in all relevant sections that the results are based on preindustrial simulations.

Though, we believe that insights obtained from water tracers in this study through a preindustrial climate simulation is valuable for the knowledge gap mentioned in the introduction.

Minor comments:

1) "novel" in the title is a bit unspecific, could be more precise

Response: Thank you. It is changed to 'innovative'.

2) L. 1: "for gaining insights into **past and future** polar, and global changes"

Response: Changed.

3) L. 1: "changes" in what exactly? Environmental changes?

Response: Yes, changed to "environmental changes".

4) L. 7-10 and results section about Fig. 4: "The tendency of poleward vapour transport to follow moist isentropes means that central Antarctic precipitation is sourced from more equatorward (distant) sources via elevated transport pathways than coastal Antarctic precipitation. We find however this tendency breaks down in the lower troposphere, likely due to diabatic cooling." I find this analysis based on Fig. 4 interesting but also very puzzling. Fig. 4 shows a zonally averaged mass-weighted vertical cross section of the source latitude of water vapour. This Fig. is discussed in the result section in a moist isentropic framework, from which we would expect moisture from a given latitude to follow the moist isentrope corresponding to the surface equivalent potential temperature from that latitude. I am a not so sure about what Fig. 4 tells us exactly:

To me it seems like the moist isentropic framework is a very crude approximation to the typical transport pathways and doesn't provide more than a justification for the fact that precipitation falling on the plateau tends to come from further equatorward than coastal precipitation. Similarly, it most likely explains why precipitation in the warm season comes from further equatorward than in the cold season. But other than that, when looking at Fig. 4, I see mainly deviations from the moist isentropic framework. On the Antarctic plateau the highest elevations don't pool their moisture from the most equatorward/warmest sources. And even more generally, in the upper troposphere, I see substantial deviations of the steepness of the source latitude contours from the moist isentropes. Of course, we expect that because in these dry upper tropospheric regions water vapour can be substantially older and make the distribution of source latitudes much wider. Could the authors maybe provide a weighted mean standard deviation or interquartile range of the source latitude in addition to the mean? This would provide a way of characterising the widths of the source distributions. I would assume that it is widest in the center of the storm track but maybe I am wrong.

→ Thus, in short, from this analysis, I see mainly deviations from the moist isentropic poleward moisture transport framework, rather than agreement with it. I think this aspect ought to be discussed more in depth along with Fig. 4 (both panels). And, in particular, if diabatic processes are invoked for explaining these deviations, then why not name, which one the authors think could play a role?

Response: We appreciate this comment, and we fully agree with the referee. We rephrased the following sentences to stress the deviations.

The part in the abstract is changed to "Central Antarctic precipitation is sourced from more equatorward (distant) sources via elevated transport pathways than coastal Antarctic precipitation. This has been attributed to a moist isentropic framework, i.e. poleward vapour transport tends to follow constant equivalent potential temperature. However, we find notable deviations from this tendency especially in the lower troposphere, likely due to radiative cooling."

The part In the result section is changed to "This pattern has been attributed to a moist isentropic framework (Pauluis et al., 2010; Bailey et al., 2019; Wang et al., 2020), which suggests that poleward moisture transport follows moist isentropes, i.e. contours of equivalent potential temperature. However, we find notable deviations from this framework especially in the lower troposphere (Fig. 4), as moisture transport pathways intersect moist isentropes. This might be expected due to the radiative cooling effects of water vapour in the troposphere (Manabe and Strickler, 1964)."

We checked the annual standard deviation of zonal mean moisture source latitude of atmospheric humidity as in the following figure. The magnitude is generally small and is larger in polar regions.

[Figure]

5) L. 14: "wind10": this variable is not yet defined in the abstract, please be specific at this stage. I also think that it is not such an elegant variable name for the main text.

Response: wind10 is defined in L. 7. Yes, but we might not easily find a better name for it.

6) L. 63: "issues with the **Lagrangian** identification of precipitation events solely using thresholds in specific humidity changes". Yes, true, but reproducing precipitation events at the right place and at the right time is tricky as well for ECHAM6. Actually, as long as the precipitation statistics are faithfully reproduced over the ice sheet over the time period considered, this aspect of precipitation event representation is irrelevant in the context of a free running simulation. So, I am not sure this is a fair point to make at this stage. I would simply say that the Eulerian method presented in this study is complementary to the Lagrangian one and offers an elegant online diagnostic of the moisture sources of Antarctic precipitation.

Response: Thank you. We deleted that statement and added in the next paragraph: "This Eulerian method is complementary to the Lagrangian one and offers an elegant online diagnostic of moisture sources."

7) L. 81: "revised definitions of Heavy Precipitation": revised compared to what? I would remove "revised". Maybe also think about not using any abbreviation (and not necessarily capital letters) for heavy and light precipitation because there are already many abbreviations used in this paper.

Response: We removed the word "revised" and the abbreviations for heavy and light precipitation.

8) L. 92: "the EAIS may be slightly too low in elevations" Why may? Some "valleys" may be missing too in the topography at this coarse resolution. Do you mean that the EAIS is on average too low?

Response: We wanted to express that the peak elevation over EAIS is lower in the simulation than in nature. We removed this phrase to avoid ambiguity.

9) L. 102: is $q^{near\_sfc}$ the $q$ at the lowest model level?

Response: Yes. We modified the variable description accordingly.

10) L. 120: to characterise the near-surface humidity gradient $q_{near\_surface}$-$q_{SST}$ or $RH_{SST}$ would be more effective, see Aemisegger and Papritz et al. 2018
Aemisegger, F., and L. Papritz, 2018: A Climatology of Strong Large-Scale Ocean Evaporation Events. Part I: Identification, Global Distribution, and Associated Climate Conditions. J. Climate, 31, 7287–7312, https://doi.org/10.1175/JCLI-D-17-0591.1.

Response: Thanks for this comment. This is indeed one of our current research questions: whether it has added value to trace $RH_{sst}$ in the study of water isotopes.

11)  L. 125: "moisture parcels" -> moisture in air parcels?

Response: removed 'parcels'.

12)  L. 141: "The difference..." in what? In the chosen definition of LP and HP?

Response: modified as "These definitions".

13)  Section 2.3: Do I understand it correctly that the LP and HP definitions are chosen such that these categories contribute to a significant share of the mass balance? Just to be symmetric in the information given about the two categories: how much does the HP category contribute to total precipitation? (i.e. what is the top-10%-precipitation-days' share of total precipitation?).

Response: Yes, while light precipitation contributes to 10% of total precipitation by definition, heavy precipitation contributes to 30 to 70% of total precipitation as shown in Fig. B4.

14)  L. 154: "on the low side" -> lower than the reconstruction, "high across some coastal areas" -> larger than in the reconstruction

Response: changed.

15)  L. 157: I would say that ECHAM6 clearly shows a larger interannual variability (2x larger than the reconstruction): is this expected given the temporal resolution of the ice core data?

Response: Yes, there is larger interannual variability in the simulation. We are not sure whether it is because the model simulates too large variability or the accumulation product based on ice core data did not capture enough variability. The ice core data is annually resolved, but diffusion in the ice might smooth out the variability. We modified the sentence to: "Interannual variability, measured as the percentage of annual standard deviation to the annual mean, is slightly higher in the ECHAM6 simulation (~20%) than in the Medley dataset (~10%)." The investigation of this variability difference is out of the scope of this study.

16) L. 160: mention the ERA5 period with which you compared your preindustrial simulation. The warm season precipitation (NDJFMA) is quite substantially higher in ERA5, is this an effect of the slightly warmer Southern Hemisphere atmosphere in the period 1979-2021?

Response: Thanks, this can be a very valid explanation. We did find that there is a significant increasing trend in Antarctic precipitation in a CMIP6 historical simulation using AWI-ESM, which uses ECHAM6 as the atmospheric component. So we are not surprised that our preindustrial simulation using ECHAM6 shows less precipitation than ERA5.

17)  L. 169-170: make clear that this is in the annual mean.

Response: added.

18) L. 170: The share of moisture sourced from sublimation over Antarctica probably depends on the parametrisation of the surface sublimation flux (e.g. Gerber et al. 2023). Could it be that the regions affected by very high sublimation fluxes with e.g. strong katabatic winds and blowing snow sublimation tend to feed cold air outbreaks and contribute more to precipitation over the ocean? Or are the sublimation fluxes over Antarctica just so small compared to the available moisture in the atmosphere? Gerber, F., Sharma, V., & Lehning, M. (2023). CRYOWRF—Model evaluation and the effect of blowing snow on the Antarctic surface mass balance. Journal of Geophysical Research: Atmospheres, 128, e2022JD037744, https://doi.org/10.1029/ 2022JD037744.

Response: We fully agree that the contribution to total precipitation from the Antarctic ice sheet depends on the parameterisation of surface sublimation fluxes. And we find notable differences in this contribution from simulations of ECHAM6 and the Unified Model (UM) using both the same water tracing diagnostics. This will be discussed in a future paper from our group. We added one sentence: "The magnitude of continental recycling depends on the parameterisation of surface sublimation fluxes (Gerber et al., 2023) and thus requires further investigation, e.g. inter-model comparisons or sensitivity tests of surface schemes."

We only find notable contributions from Antarctica to oceanic precipitation over the Ross ice shelf. Again, it depends on the parameterisation.

19) L. 179: does this finding about the precipitation coming from the open ocean south of 50°S also relate to the fact that the ocean surface of this oceanic region is larger around the AP and WAIS than for the EAIS? Does it have to do with the sea ice extent in the different basins? Does the steeper topography of the EAIS play a role (forcing the rain out of the humidity sourced from the South Atlantic south of 50°S along the EAIS slopes). Also, in terms of dynamical drivers of this share of precipitation from south of 50°S: it is really interesting to note that the Dronning Maud land receives the most equatorward moisture of whole Antarctica (even though the highest elevation of the Plateau lies much further to the southeast). This might be linked to the spiral shaped form of the Southern Ocean storm track. In the South Atlantic many extratropical cyclone genesis points are climatologically located relatively far North compared to the South Pacific (see Wernli and Schwierz, 2006). Cyclones likely play a key role in poleward moisture transport in this region.

Wernli, H., and C. Schwierz, 2006: Surface Cyclones in the ERA-40 Dataset (1958– 2001). Part I: Novel Identification Method and Global Climatology. J. Atmos. Sci., 63, 2486–2507, https://doi.org/10.1175/JAS3766.1.

Response: We very much appreciate these points. Those are valid hypotheses that different contributions to EAIS/WAIS/AP precipitation from the ocean south of 50°S can be related to the geographic size of the ocean regions, sea ice extent, and topography (higher in EAIS and thus less moisture from nearby oceans).

We do think extratropical cyclones play an important role in poleward moisture transport. It could be a very promising study to combine climatology of cyclones and moisture source diagnostics, which is unfortunately not in the plan of our current study. There are a few other findings in this study that could potentially be better understood with a better knowledge of cyclone climatology: Pacific and Indian oceans contribute more than twice to Antarctic precipitation than the Atlantic ocean, while their geographic sizes are not twice larger; Antarctic precipitation is sourced from windier conditions than usual (~2m/s), which might be linked to cyclone activities.

20) L. 186 & L. 194: "tends to take an elevated path" -> "has to rise to higher altitudes"? An elevated pathway nearly sounds as if it would travel in the upper troposphere for a very long time before raining out over the Antarctic ice sheet.

Response: Agreed and changed.

21) L. 189: "so" -> "therefore"

Response: changed.

22) L. 191: "approximating a moist adiabatic poleward ascent", this tendency is discussed already in Stohl and Sodemann, 2009, albeit with isentropes (not moist isentropes) Stohl,A.,and Sodemann,H.(2010), Characteristicsofatmospherictransportintothe Antarctic troposphere, J. Geophys. Res., 115, D02305, doi:10.1029/2009JD012536.

Response: Thanks, changed.

23) L. 191: "this tendency to follow contours of equivalent potential temperature breaks down in the lower troposphere" -> tendency to follow moist isentropes?

Response: This sentence is removed as in the response to the first comment.

24) L. 191: about the "break-down" of the moist isentropic approximation of poleward transport: Which diabatic cooling mechanism do the authors think plays a role? Could diabatic heating especially in the upper troposphere also explain part of the observed deviations from the moist isentropic framework?

Response: The diabatic cooling mainly results from radiative cooling effects of water vapour (Manabe and Strickler, 1964). Yes, diabatic heating is most obvious in tropical mid-to-upper troposphere. Although equivalent potential temperature is conserved during vapour condensation for a given air parcel, latent heat release might still heat remaining moisture to cross moist isentropes.

25) L. 197: as mentioned in my general comment: just make sure that it stays in the mind of the reader that this study covers another time period.

Response: added: "though their study was for present-day climate rather than preindustrial climate as in our study."

26) L. 201: most equatorward sources for DJF: that is surprising because of the minimal sea ice extent and the slightly weaker jet and storm track in summer. Is this simply due to the warmer atmosphere on average, leading to higher humidity contents and longer transport distances? And as you write flatter isentropes "giving access" to more equatorward sources?

Response: Thanks for this comment. We were also surprised by this result. As sea ice is at minimum during austral summer DJF, we would expect the opposite to be true. We thought of two mechanisms as written in the text: 1) flatter moist isentropes in DJF (not convincing enough as there are significant deviations from the moist isentropic framework, so removed now); 2) weaker westerlies in DJF. The second hypothesis is partly supported by our results in Section 3.5.

In DJF, the atmosphere is warmer, humidity is higher, and oceanic evaporation is lower, which means longer residence time of moisture. However, we are not sure whether it means longer transport distance.

27) L. 203: milder->flatter

Response: changed.

28) L. 209: The fact that source longitude shows the largest interannual variability is interesting and shows that the strength of the westerlies and the storm track dynamics is likely important for modulating the moisture source properties.

Response: Thank you for this point. We fully agree and added the following: "It suggests that the strength of southern westerlies and the storm track dynamics are likely important for modulating the moisture source properties."

29) L. 211: "source latitude and longitude"

Response: Changed.

30) L. 214: "lies within the range of estimates from the literature"

Response: Changed.

31) L. 220-227: Interesting! This corresponds to the range of wind speeds associated with events of strong ocean evaporation in the Southern Ocean as discussed by Aemisegger and Papritz, 2018. Extratropical cyclones and trailing fronts were identified as key weather systems with which enhanced ocean evaporation is associated.

Response: It is really nice to know this relevant study. Based on Eq. 1 it can be expected that if wind speed increases while other variables staying the same, evaporation will increase linearly with wind speed. It might be relevant to project how evaporation will change under climate change and how it impacts Antarctic precipitation.

32) L. 226: "other forms of storms" what is meant here: tropical cyclones or polar lows? I think if you write "extratropical cyclones propagating along the Southern Ocean storm track" you also include subgroups such as mesocyclones or polar lows.

Response: Thank you for pointing it out. We changed it as suggested: "extratropical cyclones propagating along the Southern Ocean storm track".

33) L. 230: what does "evaporation processes will cause some decoupling of moisture source properties" mean? Do you mean that synoptic-scale variability at the source causes variability in the evaporative conditions?

Response: No. We meant moisture source properties can be slightly decoupled from moisture source locations because of their impacts on evaporation.

34) L. 234: what do you mean by "wind10 at source" -> climatological wind 10 at the source?

Response: Yes, changed.

35) L. 239: in the previous two paragraphs you just discussed **dynamic** not thermodynamic controls on moisture availability for Antarctic precipitation.

Response: Yes, we changed it to "dynamic control".

36) L. 256: A case study discussing this suppression of ocean evaporation during polarward moisture transport in warm sectors of extratropical cyclones is Thurnherr and Aemisegger, 2022.

Thurnherr, I. and Aemisegger, F.: Disentangling the impact of air–sea interaction and boundary layer cloud formation on stable water isotope signals in the warm sector of a Southern Ocean cyclone, Atmos. Chem. Phys., 22, 10353–10373, https://doi.org/10.5194/acp-22-10353-2022, 2022.

Response: Thank you for pointing to this nice study as further supporting evidence. It is cited now.

37) L. 282-285: The impact of the SAM on the storm track dynamics and shifts in strong ocean evaporation patterns as well as near surface conditions is discussed in Aemisegger and Papritz, 2018.

Response: Thank you for letting us know. It is cited.

38) L. 291: "These results quantify the degree to which poleward moisture fluxes are associated with the SAM" -> are modulated by the SAM? And maybe instead of fluxes rather use poleward moisture transport?

Response: Thanks, changed.

39) Fig. 10: the stippling is very difficult to see

Response: Thank you for pointing this out. We enlarged the stippling in all related figures.

40) L. 300: "SAM states exert controls" -> "the SAM impacts"

Response: Changed.

41) L. 310: "limits on" -> "of"

Response: Changed.

42) L. 328: "yields a more precise value compared to previous method" -> explain why explicitly

Response: This sentence is removed here and the pros and cons of our method compared to previous ones are discussed in detail in subsection 2.2.1.

43) L. 330-332: this is a bit obscure to me. True in close approximation for SST but not necessarily for rh2m and the wind speed.

Response: Yes, this is more valid for SST, but also valid for rh2m and wind10. Correlation analyses indicate high correlation between source latitude and source properties (highest for SST and lowest for wind10 as shown below for EDC as an example). Predominant meridional gradients (Fig. B2) of these variables could partly explain these correlations, though we notice a meridional maximum of wind10 at around 50 degree south. The decoupling between source wind10 and source latitude is then explained in the next paragraph.

[Figure]

44) L. 355: "other types of tracers in numerical systems" -> what do you mean by this?

Response: We modified it to be clearer: "Finally, we note the new scaled-flux tracing approach is applicable not only to water tracers in atmospheric GCMs but also to other types of tracers, e.g. aerosol tracers, in numerical systems".

45) L. 356: "the full potential of our water tracing diagnostics is yet to be identified" -> This last sentence is a bit unspecific for the end of such a nice paper.

Response: Thank you. This sentence is removed.

46) Appendix A: very nice and interesting documentation. I just get lost at line 385. What does the index *i* represent and what do you mean when you write "by summing up all vapour contributions in a grid box, we obtain:"? Do you sum up over a number of grid points in a given region? In equation A4 some variables have an index *i* and others not. I didn't get why. And in A1 *t*, $\lambda$, $\phi$ have an index *i*, but in the next lines not. Sorry to be picky, but I really would like to understand this Appendix.

Response: Thank you, this is a nice point to improve the manuscript. We added the following to explain "i": "For any infinitesimal evaporative flux i from the open ocean". Indeed, it is an abstraction of infinitesimal moisture parcels in the model.

By "By summing up all vapour contributions in a grid box, we obtain", we meant summing up all infinitesimal moisture parcels in a grid box, which represents all moisture in a grid box.

In A4, if t, $\lambda$, $\phi$ have an index i, they represent the time, latitude and longitude associated with the evaporation flux i; if t, $\lambda$, $\phi$ do not have an index i, they represent any subsequent time, latitude and longitude. The same for A1.

47) Appendix A: I like your comparison with the predefined-region water tracing method described at lines 410ff a lot. To me this is a highly effective way to show the advantages of the scaled-flux approach and I would find it very helpful to have a short version of the results from this comparison in the main text.

Response: We appreciate this comment and we added this figure and associated text to a new subsection 2.2.1, also as a response to comments of the second referee.

---

## Author Comment (AC2)

**Responses to comments of Referee #2**

We are thankful for these valuable comments. These comments are very constructive in improving the manuscript.

This paper presents an analysis of the moisture sources of Antarctica based on a pre-industrial climate model simulation. The authors present a combination of tagged water tracers and source property tracers, which is new for this climate model. Overall, the results from this study are mostly consistent with and confirm previous model results. I find this study overall interesting and valuable, as the combination of these tracer diagnostics has not been applied to Antarctica before. **However, there are several aspects, including the connection to literature, the structure, selection and description of the material, and the claimed significance of the findings that require major revisions**, as detailed in the comments below.

We modified the manuscript based on these suggestions. Specifically, we improved the connection to previous literature; we added one subsection to evaluate the scaled-flux water tracer method; and we removed four figures in the appendix and reordered other relevant figures.

**Major comments:**

1. There are several cases where the connection to previous literature is not made sufficiently clear. The findings here appear to be very consistent with previous results. I find it remarkable that different methods end up with such similar numbers. It should be stated more clearly that your results confirm previous work, thereby also maintaining and strengthening studies that have been building on similar numbers for source region contributions. This is an important conclusion that needs to appear in the abstract, the results, and the conclusions, with suitable referencing.

   Response: We appreciate this comment, and we mentioned this consistency with existing literature in the abstract, results, and conclusion in the revised manuscript.

2. The authors claim a novel and in particular precise method is being used. However, there is almost no material that would underpin the validity of this claim, apart from figure B11 in the appendix, which in itself is not very convincing. Additional quantitative support for the equivalence of the method could for example originate from a mass-weighted mean of a setup of both latitude and longitude boxes. A section discussing the performance of the new method should be added.

   Response: Thanks, we added a new section 2.2.1 to evaluate and discuss the performance of the new method. Please also see our response below to your specific comment on line 121 from the original manuscript.

3. In relation to that, it is not clear from the results when either one of the two tracer approaches is used, and when they are used in combination. It would help the reader to clarify the connection of findings to either of the two methods, and highlight the novelty and additional value of the authors' approach.

   Response: Thank you for this point. We added this information in all relevant figures and text parts.

4. **The authors state the results are more precise, both in the abstract and elsewhere**. I think this statement is not entirely correct, since only weighted mean values are transferred to the target location. The uncertainty range or spread of source region properties at every location is simply not represented any more due to the averaging of source information. Maybe I misunderstood how the precision is meant, but in any case this is a topic that needs to be discussed critically.

   Response: We agree with the reviewer on this point. As stated above, we added a new subsection 2.2.1 on this topic.

5. **Several key figures are not introduced and described properly, only a general takeaway is given.** This makes it difficult for the reader to follow the argumentation, and to go back and forth between text and figures. See detailed comments.

   Response: We ensured that all figures and panels are properly referenced now.

6. The selection of material and its placement need improvement. There are too many figures and figure panels in the appendix, some of which are extensively described in the main text, others only stand with a reference and figure caption. The authors should carefully consider which figure panels are central to their results and needed to underpin their findings, and which can be removed. At the moment, there is a lot of figure material that the reader is left with on their own. **As a general recommendation, remove all those panels that are not relevant for the flow of the argumentation, and include those that are discussed in the text in the main manuscript.**

   Response: Thank you. We removed four figures in the appendix (Fig. B4, B7, B9, and B10 in previous version) and replaced one into the main text.

7. Consider changing some aspects of the writing style. For example, section title 3.2 to 3.5 are formulated as questions, which I find not entirely fitting as a title. **There are some paragraphs that serve as table of content for sections to come, which perturb the flow of reading, rather than being helpful.** There are also a few casual formulations, such as referring to the simulation as "our"

simulation. As a reader, I wonder why the authors put such strong ownership into a study object.

Response: We appreciate these comments. Titles of section 3.2 to 3.5 are changed. Paragraphs serving as table of content are removed. We also removed all statements that might show such ownerships.

8. The significance of the wind speed differences at the source regions in general, and when they contribute to Antarctic precipitation, is not clear. There is also a claim of thermodynamic evaporation effects in the abstract, while wind would usually be considered as a result of pressure gradients and thus atmospheric dynamics rather than thermodynamics. Since the authors present this as a major finding, the overal reasoning and significance of this result should be presented more clearly.

    Response: Thanks, we rephrased wind speed as a dynamic control. The reasoning of stressing the significance of the wind speed differences is rephrased, see responses to comments on L. 14-16 below.

9. The distinction between high and low precipitation events could be shortened considerably.

    Response: Thanks. It is shortened in Section 2.3.

10. **There is overall too little mentioning of limitations and critical evaluation of the method.** For example, the tracer method does not allow to reconstruct source footprints, and it only transfers a weighted average of the source properties, not their original range of values.

    Response: Thank you for these valuable points. These are added to the new section 2.2.1.

**Minor comments**

L. 4: precise: maybe rather say 'detailed'? There is no complete uncertainty range available from this method, but that does not make it more precise.

Response: Thank you. It is changed to "detailed".

**L. 14-16: Moreover, ...: not clear that this is a result which should be highlighted in the abstract. What is the concrete relevance of this finding?**

Response: Thanks for pointing this out. We do think this is a relevant finding and should be highlighted in the abstract.

This finding highlights the impact of wind speed on evaporation and moisture supply for the atmospheric water cycle. It provides a new perspective to think about changes in

atmospheric water cycle under climate changes. In addition to increased moisture holding capacity of a warming atmosphere, we suggest that changes in moisture supply through oceanic evaporation also play a role in the intensification of global water cycle.

Indeed this was also pointed out in the paper by Sodemann and Stohl, (2009): "During both seasons, moisture sources for Antarctic precipitation are distributed annually in the Southern Ocean, with distinct maxima in the Indian Ocean sector at about 40°S. This corresponds to a maximum of **surface wind energy** associated with fronts that are related to maxima in cyclone density and baroclinicity further south [Simmonds et al., 2003]." And "the evaporation contribution maxima in the Indian Ocean and the Pacific sector are associated with maxima in latent heat flux and **surface wind velocity**."

Here we show additional evidence relating wind speed and oceanic evaporation.

L. 30-33: The topic sentence here is on diamond dust, but then the discussion switches immediately to marine air intrusion events. I think the topic sentence should be about precipitation in general instead.

Response: We appreciate this comment. We added one topic sentence: "Antarctic precipitation can manifest in various forms."

**L. 43 onward: It has long been known that there are thermodynamic limitations in how vapour can reach low-lying and higher areas of Antarctica, see for example Fig. 3 in Stohl and Sodemann (2010). This would be helpful to include here, since the discussion comes back to this aspect later in the manuscript.**

Response: Thanks, it is included now.

L. 48: "dominate" - this seems to contradict the statements above in line 32.

Response: Changed to "contribute significantly to".

L. 57: "moisture flux tracks" - rephrase as "moisture transport paths"

Response: Changed.

L. 59: "most commonly" - I think this can be debated, different tools have dominated in different time periods, and source region tracers came definitely first

Response: Yes, We changed it to "One of the widely applied tools".

L. 60 onward: I think it would be useful here to state what was found in these studies, since you come back later to this, and compare. For example, what were the main source regions, the average latitude, pattern.

Another aspect that would be fair to bring up here is that these Lagrangian studies allow to obtain maps of the source regions at spatial detail, which does not seem to be available easily from either of the methods applied here.

Response: Thank you for this comment. We fully agree and added the following:

"Based on a meteorological analysis dataset from October 1999 to April 2005, Sodemann and Stohl (2009) diagnosed moisture sources and sinks through changes in specific humidity along transport pathways of air parcels. While only ~90% of total precipitation could be attributed to specific sources with 20-day backward trajectories, annual moisture source latitudes of precipitation over Antarctic Plateau were estimated to be 45 to 40° S. Moisture source longitudes were generally located at 20 to 60° to the west of precipitation locations. They also pointed out seasonal variations in moisture source latitudes of Antarctic precipitation, which are related to Antarctic topography, sea ice, baroclinicity, and mid-latitude land-sea distributions."

It is indeed possible to get such maps of moisture source contributions to precipitation at a specific location or whole of Antarctica from the water tracing methods, though in an approximated way. The following figure shows contributions of each grid box (1° * 1°) to annual mean precipitation at Dome C, based on the scaled-flux water tracing method. To obtain these contributions, we attribute daily precipitation at Dome C to a grid box where the moisture source latitude and longitude are located at (i.e. projecting daily precipitation backwards to evaporative sources). Then by summing up over 60-year period, we get these relative contributions. This is not the same as backward projection from Lagrangian trajectories, as there is a mixing of different moisture sources (so some precipitation is attributed to land masses).

[Figure]

L. 69: "While..." I do not understand where this information belongs. Rephrase?

Response: Thank you. We found that this sentence is confusing for many readers and does not provide much valuable information, so it is removed.

L. 76: "For the first time..." - I do not think this is a valid claim. There have been many studies before of Antarctic precipitation origin, both from climate model tracers and with trajectory approaches. As all other approaches, your methods have their limitations. I recommend to moderate this statement.

Response: Fully agree and removed.

L. 80-83: This table of content sentences appear unnecessary and can be removed.

Response: removed.

L. 93: Please state the exact elevation difference

Response: Thanks for this comment. This is also mentioned by the first referee in the 8[th] comment. We removed this statement in that response.

L. 101: This is a bit of a confusing statement. Wind at the surface is by definition zero. The equation also just states the wind at the lowest surface level, rather than a vertial wind gradient.

Response: Yes, we agree. We changed the wording to "the wind speed at the lowest model level".

L. 121: Since this is a new implementation, some demonstration of the performance and evaluation should be given in the main manuscript. The comparison shown in Fig. B11 is also not very convincing, and only considers the "easy" case of latitude boxes. How well does this compare to, for example, in a setup with longitude boxes?

Response: We agree that such a suggested demonstration of the performance of our approach in the main manuscript would be helpful. Thus, we have added a subsection 2.2.1 to evaluate the scaled-flux water tracing method.

The comparison of a latitude vs. longitude setup is a very good question. We did similar analysis as Fig. B11 for all other source properties, as shown in the following figure for source longitude. The first row shows moisture source longitude of annual, DJF and JJA precipitation based on the scaled flux water tracing method. The second row shows the differences between the first row and moisture source longitude based on the prescribed-region water tracing method, which divides the globe into 20°-wide longitude bins. The third row is similar as the second row, but with 10°-wide longitude bins. A similar figure for source SST is attached below as well. The case for SST is slightly more complicated, because the source region varies at every time step with the variations of SST.

Since the two different tracing methods provide very similar results, and prescribed-region water tracers with finer bins provide results in closer agreement to scaled-flux water tracers as prescribed-region tracers with coarser bins, we are confident that the scaled flux tracers correctly reflect moisture source regions and properties in ECHAM6.

[Figure]

L. 125: The direct source-sink distance with underestimate the transport distance.

Response: True, we changed it to "Note that this geographical distance is smaller than the actual transport distance of the moisture."

L. 135: **0.002 mm day-1 is a much smaller number than can be measured in reality.** How useful is it to define this threshold? How sensitive are your results to

the choice of this threshold? This is an example for factors that contribute to the uncertainty of your results and conclusions, and should be discussed openly.

Response: We am not sure about the lowest precipitation amount that can be measured in the field, but we found daily precipitation amount lower than 0.002 mm/day observed over Dome C in the supplementary file of Stenni et al. (2016). Also note that this threshold is applied for a grid-box mean precipitation amount, rather than a specific site.

While Turner et al. (2019) used a larger threshold of 0.02 mm/day to define a precipitation day, we find it necessary to use a smaller threshold for our case. As shown in the following figure (a) and (b), days with a precipitation rate lower than 0.02 mm/day can contribute up to 10% of total precipitation over the Antarctic Plateau in both (a) ERA5 and (b) ECHAM6. So it might be problematic to exclude them.

We made this choice to make the analysis more robust, though our results are not very sensitive to the choice. As shown in Figure (c) and (d), the differences between moisture source latitude of heavy precipitation and the rest of precipitation are similar for a threshold of (c) 0.002 mm/day and (d) 0.02 mm/day. Though, the choice of 0.002 mm/day makes the results more conservative (slightly smaller magnitude of anomalies in c than d), because a smaller threshold means more days are included as heavy precipitation by definition, which leads to smaller differences to the rest of precipitation.

[Figure]

(a) ERA5:                                                    (b) ECHAM6:

[Figure]

(c) 0.002          Source latitude anomalies [°]          (d)0.02          Source latitude anomalies [°]

L. 140: It is not clear whether this percentile is a local choice at every grid point, or for the overall precipitation

Response: It is for the overall precipitation. The statement is now modified: "This is because a definition of light precipitation as the 10% lowest precipitation days would contribute to less than 0.3% of total Antarctic precipitation, and less than 1.1% of total precipitation at individual grid boxes."

L. 147-151: This table of contents section is not needed and can be removed

Response: Thanks, removed.

L. 153: Figure 1 is not introduced and described properly, only a general takeaway is given. This makes it difficult for the reader to examine the figure. The same applies to many other figures in the manuscript. Individual figure panels need to be referred to in the text.

Response: Thank you for this comment. We added the reference to individual figure panels where relevant. We will also make sure other figures and subpanels are properly introduced.

L. 160: Fig. B3 and B4 are discussed in the text, so it would be natural to include those with Fig. 1 as sub-panels. Fig. B4 would come logically before Fig. B3.

Response: Thanks. We removed Fig. B4 as a response to your major comment #6. Though Fig. B3 shows that seasonality of Antarctic precipitation is captured in ECHAM6 preindustrial simulation compared to ERA5 and it would be nice to show this figure in the paper, we prefer to put it in the appendix to limit the amount of figures in the main text to a minimum to stress the findings.

L. 167: "suggests that both": I find this conclusion too vague. Can this be documented concretely? How do we know the information in Fig. B5 is sufficient as a basis for further analysis?

Response: Yes. We added the following to the main text: "Simulation results deviate less than one standard deviation from ERA5 for both MSLP at 40° S and 65° S (Fig. B4a) and for their differences (Fig. Bb). Root mean squared errors between simulated and assimilated MSLP at 40° S and 65° S, and their differences, are 1.0, 1.4, and 1.5 hPa, respectively."

Section 3.2 does not work well. **I understood only much later that you use here the source region tracer to extract the provided information** (if I understood correctly). The information in Fig. 2 and 3 is not introduced and described well enough to capture the information easily. Can Fig. 2 and 3 be combined into one figure? Fig. 3 would also be more logical to look at before Fig. 2.

Response: Thank you for bringing this up. We added in the figure caption whether the moisture source information is from prescribed-region or scaled-flux water tracers. We also added the related information in the text.

We also switch the position of Figure 3 and 2 now. Though we describe Fig 2 and 3 together, it might be better not to combine it to avoid confusion. We added a reference to each subpanel to further strengthen the link between text and figures.

L. 176: Where in Fig. 3 can these percent contributions be seen?

Response: We removed the reference to Fig. 3 here to avoid confusion. The total contributions from AP/EAIS/WAIS are not displayed in the figures.

L. 177: the maximum -> it's maximum

Response: changed

Figure 4: This is an interesting figure, but I am not sure I interpret it correctly, and it is only discussed briefly. Is this figure showing all moisture in the atmosphere, or only such that contributes to precipitation in Antarctica? Has this result been obtained with the source region tracer, or the source property tracer?

Response: This figure shows all moisture in the atmosphere originated from the open ocean. We also added in the figure caption: "Moisture source latitude information is derived from the scaled-flux water tracers."

L. 186: In this context, there are several studies that can be referred to, including Stohl and Sodemann, 2010 and Terpstra et al., 2021.

Response: We added the suggested citations.

L. 203: milder -> warmer

Response: Thanks. This sentence is deleted based on responses to comment 26 of the 1$^{st}$ referee.

L. 204: "may also play a role": can this speculation be backed up in some way?

Response: We rephrased it: "We propose that weaker westerlies in DJF compared to JJA, induced by smaller meridional thermal gradients, may promote equatorward shifted moisture sources (see Sec 3.5 for details)."

This speculation is partly backed up through analysis in Sec 3.5, where we find that negative SAM phases associated with weaker westerlies are linked to equatorward shifted moisture sources.

L. 207: "We speculate that...": can this be investigated further in the light of previous studies? Sodemann and Stohl (2009), their Fig. 1 and 2, have maps that can be directly compared to your results.

Response: Thanks, we added the following sentence: "This pattern is also observed in the results of Sodemann and Stohl (2009) for DJF precipitation (see their Fig. 2c)."  and  "This would need to be investigated through Lagrangian moisture trajectory diagnostics."

L. 209: Figure B6b seems to contain important information, should it no then be part of the main text?

Response: It is true that Figure B6 contains important information regarding seasonality of moisture sources, as described in the following text. But this figure with 30 panels might be overwhelming for many readers, so we prefer to only extract key messages from it for the main text and put the complete figure in the appendix for interested readers.

L. 211: "Having dealt with...": rephrase, find a better connection/transition.

Response: Changed to "After studying moisture source regions and locations".

L. 223-226: rephrase to provide a more direct message. What are "other forms of storms"?

Response: Changed to "extratropical cyclones propagating along the Southern Ocean storm track" based on comment 32 of the 1$^{st}$ referee.

L. 229: Eq. 1 does not contain rh2m nor SST

Response: True, we rephrased it to "Firstly, oceanic evaporation is related to wind10, rh2m, and SST (Eq. 1)."

L. 229: Hard to follow the discussion here. Can it be said more clearly what decouples from one another?

Response: Yes, we changed it to "moisture source properties can be slightly decoupled from moisture source locations because of their impacts on evaporation" based on comment 33 of the 1$^{st}$ referee.

L. 231-238: This message in this text is not very clear. What exactly is the difference between panels a and b? How do you explain the difference? And why is this a thermodynamic control? Wind speed would generally be categorized as dynamic rather than thermodynamic.

Response: We rephrased this paragraph. While Panel a is moisture source wind10 of annual mean Antarctic precipitation, Panel b is annual mean wind10 at the moisture source latitude and longitude of annual mean Antarctic precipitation. The differences indicate that evaporation occurs preferentially during higher wind speeds at oceanic grid boxes. We changed the wording also to express that wind speed is categorized as a dynamic control.

Could it be interesting to set these findings in perspective with histogram plots, and identify the percentile of evaporation events that contribute on average to Antarctic precipitation?

Response: We fully agree that it is interesting to associate strong Southern Ocean evaporation events with Antarctic precipitation, as pointed out by comment 31 of the 1st referee (Aemisegger and Papritz, 2018). While this is out of the scope of this study, we will consider it in the future with historical and future simulations.

L. 241-246: It is not clear what information should be picked up from this paragraph, and what is the conclusion. The information is provided in an appendix figure instead of the main text, which makes it very difficult to see the relevant information together with the text. Either this paragraph and the appendix figure should be removed, or selected panels be shown in the main text to make the information accessible.

Response: Thanks for this point. We shortened the text and combined it with the previous paragraph as below:

"Secondly, annual cycles of source latitude and properties are controlled by meridional thermal gradients, sea ice variations, and seasonal climate variations at mid-latitudes. For example, precipitation is from more southern oceans in MAM because of less sea ice than JJA (Fig. B4a3 vs. a4); precipitation is from warmer oceans in MAM due to higher SST at mid-latitudes than JJA (Fig. B4d3 vs. d4); and precipitation is from less windy regions in DJF due to weaker westerlies than JJA (Fig. B4f2 vs. f4)."

As said in the previous response, we tend not to put Fig. B4 in the main text.

L. 253: This appendix figure is not self-explaining. How does it connect to what is said in the text? Is this a necessary figure to include?

Response: This figure is removed.

L. 254: The contents of Fig. 8 are not described here, but after Fig. 9 is introduced. Reorder figures or discussion.

Response: Figures have been reordered.

L. 258: "This hypothesis...". I have the impression the word hypothesis is used wrongly here. Terpstra et al., (2021) rather state this as a finding of their case study, and you hypothesize that this can be of more general validity, which is what you test in your analysis - correct?

Response: Yes, it is changed to "This finding".

Figure 8: How important is the **0.001 mm day-1** threshold value for these results?

Response: As described in the previous response, a change in threshold from 0.02 mm/day to 0.002 mm/day does not change the results nor conclusion significantly.

Figure 9: Do all these figure panels need to be shown? Not all of them are discussed sufficiently at the moment.

Response: Yes, all figure panels are referenced, and thus we prefer to show all panels.

The positive source latitude difference being more equatorward is counter-intuitive. Can this be reversed?

Response: Thanks, this is good to know. But we are sorry that there might not be a perfect wording for everyone. We thought the stated positive difference is intuitive because the magnitude of latitude is from -90 to 0 in the Southern Hemisphere.

L. 263: I would make sense to flip numbering of Fig. 8 and 9 as they appear in the manuscript.

Response: Thanks, flipped.

L. 287: Figure 10 is not properly introduced and described, only a general conclusion is given. As a reader, I am left alone in the interpretation of Fig. 10.

Response: Thank you. Figure 10 is introduced in the 2$^{nd}$ to 5$^{th}$ paragraph of Section 3.5. Each paragraph references to an individual panel. This should cover the main message we would like to deliver with these figures.

Figure B9: If this is a valuable result figure, it should be part of the main text, otherwise left out.

Response: It is left out now.

L. 303: Posing a rethorical question in a text can be confusing to the reader. Who should answer this?

Response: Thanks. It is modified: "So, we investigated whether SAM exerts control over the frequency or intensity of heavy precipitation."

L. 305: Fig. B10 is not described in the main text or appendix. If this is valuable material to include, it needs to be described properly somewhere.

Response: Thanks. It is removed.

L. 312: "we develop the dynamic": The impact of this tracking is not shown or discussed in the manuscript, and can thus not be part of the conclusions.

Response: Thanks, removed.

L. 315: "our preindustrial": puzzled by this ownership statement.

Response: Thanks, we removed all such statements.

L. 319: "spatial patterns are...": unclear what this statement means.

Response: We changed it to: "spatial patterns of the contributions are influenced by the topography".

L. 322: Bailey et al., 2019: is this the only/most relevant study to include? There are several older studies that looked into these aspects.

Response: Thanks, we added: "Stohl and Sodemann, 2010"

L. 323: The findings here appear to be very consistent with previous results. I think that is remarkable, that different methods end up with such similar numbers. It needs to be stated clearly that your results are confirming other work, maintaining and strengthening previous work that builds on these numbers. This is an important conclusion that needs to appear in the abstract, the results, and the conclusions, with referencing.

Response: Thank you for this point. We added in the conclusion: "These results are consistent with estimates based on Lagrangian trajectories (Sodemann and Stohl, 2009), which suggests a source latitude range of 45° S to 40° S for precipitation over the Antarctic Plateau."

L. 328: "more precise value": as stated earlier, I am not convinced this is true. Your method advects mass-weighted averages, which means that part of the underlying variation is simply not visible.

Response: Thanks, we explain these points now in the new subsection 2.2.1.

L. 339: It might be useful to distinguish the HP/LP results and SAM results in two sentences or paragraphs.

Response: Yes, we agree. We split it into two paragraphs in the revised manuscript.

**References**

Stohl, A., and Sodemann, H. (2010), Characteristics of atmospheric transport into the Antarctic troposphere, J. Geophys. Res., 115, D02305, doi:10.1029/2009JD012536.
**Citation**: https://doi.org/10.5194/egusphere-2023-1041-RC2

---

## Author Response (AR2)

**Responses to comments of Referee #2 - 2nd iteration**

We appreciate the efforts of Referee #2 in reviewing the manuscript. These detailed comments have significantly improved the draft.

After reading the revised manuscript and the author reply, I have the impression that the authors have responded appropriately to the reviewer comments. This revised version of the manuscript reads much better, and is more logically organized in terms of structure and content. After these revision, a few additional items stand out more clearly now that should be further improved. Most of the improvements concern the writing, and none of these points is major, so I consider this iteration a minor revision.

Minor comments:

1. The writing in the introduction can be improved. There are a couple of statements that would benefit from a reference, and where the connection and logic are not clear or not consistent.

Response: We appreciate this comment. We modified the introduction based on the detailed comments below.

2. The comparison of source properties between different components in Sec. 2.2.1 would need a form of variability quantification to be interpretable. The differences are similarly small for wind, temperature, and humidity, and it is not fully clear why wind is discussed specifically. Is it really surprising that wind speed is the main modulator in terms of evaporation strengh, given the absence of large land regions that can supply dry air (the other component of evaporation)?

Response: To save computational resources, we only run the simulation for Fig. 1b for two years, with one-year spin-up period. So we cannot present interannual variability here. However, a lack of variability quantification or statistical tests, which might require ~30 year simulations, does not afflict our conclusion here that two methods provide close results.

We mainly discussed wind in Fig. 8 because similar analysis on moisture source latitude, SST, rh2m does not show consistent patterns over Antarctica.

Yes, it is surprising to have this finding, which is also why we stressed this point in the text. We will also investigate the impact of RH relative to SST on evaporation strength.

3. Sec. 3.3, regarding source temperatures, the results does not exactly match the literature, as is being argued. Also, how large is the variability (1 sig std)? Can some of the discrepancy be explained by a differing regional focus of previous studies? A similar quantification would be useful to assess the significance of the apparently very small differences of rh2m and wind10 in different regions that are discussed in the same paragraph.

Response: True, the literature reports a large range, while using different methods. Our results lie in the middle of the estimates. The variability of source properties is shown in Fig. B5, generally less than 1 °C for precipitation source SST over Antarctica.

The two modelling studies by Koster et al. (1992) and Delaygue et al. (2000) report numbers for the whole of Antarctica, whereas Petit et al. (1991) reports numbers for surface snow collected across Antarctica. So the differences might not result from different study regions nor time periods.

4. Limitation of model in simulating storms, precipitation are never addressed. How much do we learn about the actual precipitation in Antarctica, and how much do we learn about how the model simulates precipitation in Antarctica?

Response: As this study focused on preindustrial periods, evaluation of the model simulation against recent observations of storms or precipitation can be problematic. While we consider the coarse model resolution to be insufficient to simulate coastal precipitation, there is limited observations over Antarctic Plateau to evaluate model performance. Though, our current research collected multiple datasets of precipitation and vapour isotope observations over Antarctica to evaluate model performance using ERA5-nudged simulations.

5. The method description mentions some modifications compared to Fiorella et al., 2021, and one has to go to the appendix to learn about what these modifications are and what uncertainties that imposes. I would strongly favour mentioning specifically what is modified in comparison to Fiorella when referring to the Appendix in the main text, so that readers know what they will find there. Also, the information about a <2% mass balance error from the imposed flux limits should be stated in the main text. Clearly, such an error margin will be relevant when discussing the precipitation for different amount categories (Sec. 3.4, Fig. 10).

Response: Thank you. We added the following: "For example, we only trace moisture evaporated from the open ocean, and we use three water tracers for each source property to ensure the conservation of water masses and limit the propagation of numerical errors."

Detailed comments:

L. 7 onwards: wind10 is uncommon and not specific in terms of wind speed or wind direction. I suggest changing to the more commonly used and specific abbreviation vel10 (for wind velocity).

Response: Thank you. wind10 is replaced with vel10 throughout the manuscript.

L. 17: "first time has been quantified" the highly emphasised significance of this finding remains unclear. Is it not enough to state the finding without the "first time" claim?

Considering also that this is a statement about a model, not about nature (i.e. wind measurements).

Response: Thank you. We removed the statement 'first time'.

L. 23: add reference to 2nd sentence

Response: Thank you. We added reference to Purich and Doddridge, 2023 and Gorodetskaya et al., 2023.

L. 25: not clear what "this" refers to, rewrite

Response: Changed 'On top of this' to 'Furthermore'.

L. 37: "surprisingly little": I would disagree that surprisingly litte is known about Antarctic precipitation in general. But there are for sure important knowledge gaps when it comes to precipitation at high latitudes. Consider rephrasing by stating specifically what (relevant) knowledge gaps exist. In its present form, the statement sounds a bit like the knowledge gap is with the authors - clearly a misinterpretation to try and avoid.

Response: We appreciate this comment. We changed this sentence to "our understanding of thermodynamic and dynamic factors driving Antarctic precipitation remains limited."

L. 39 onward: The sentences in this paragraph lack logical connection and stringency. Statements such as "tend to occur", "is conducive for" and "occur alongside" create a strange disconnect (as if coincidental) between dynamically deeply connected and interrelated aspects of the atmospheric flow. Please rephrase.

Response: Thank you for this comment. We changed it to: "Marine air intrusions are efficient at transporting moist and warm air from subtropics to Antarctica (Schlosser et al., 2010; Dittmann et al., 2016). The intrusions generally occur alongside strong meridional flow during planetary wave amplification (Adusumilli et al., 2021; Noone et al., 1999; Massom et al., 2004), sometimes in the form of atmospheric rivers (Gorodetskaya et al., 2014; Wille et al., 2021).

L. 46: "are mainly known" - I don't think this is a correct statement, both aspects of these precipitation events are well documented in literature

Response: Yes, we agree. This sentence is changed to "The marine air intrusions play a major role in heavy precipitation events at both coastal locations and Antarctic interior (Genthon et al., 1998; Gorodetskaya et al., 2014; Stohl and Sodemann, 2010)".

L. 48: "is useful for predicting" this could potentially be the case, but I miss the supporting evidence for this statement in the way that is written here. It is also unclear why you raise this

point here, would this not be more something for the implication of the study?

Response: Thank you. We removed this statement here.

L. 59: "it is not yet clear how SAM variations...": this statement contradicts the available knowledge about how SAM influences precipitation in the previous sentence. Maybe instead highlight specific aspects where additional knowledge gaps exist?

Response: Thanks, we changed it to "To project Antarctic precipitation changes, it is important to understand how SAM impacts moisture transport paths."

L. 125: consider simplifying this equation and writing by using simpler but equally accepted/established symbols such as q_s, q_2m

Response: Thank you, we simplified it to q_s and q.

L. 138 onwards: "quite close", "very similar": these expressions express a subjective judgement. Can this be phrased more objectively. Adding a small table for all assessed variables would be useful.

Response: We appreciate this comment. We removed these subjective statements. We only wrote the differences in source latitude from two methods in the text, and we consider it to be enough to show the confidence in the accuracy of two methods. Therefore, we do not include a small table to disturb the flow.

L. 140: clarify how this bias is computed, what is the reference?

Response: Here "bias" refers to the major deviations in results of prescribed-region water tracers from those of scaled-flux water tracers. We do not have an " absolutely accurate" reference, and the statement "bias" is indeed our judgement. We consider it as bias for two reasons: 1) The bias is inherent in the assumption that moisture source latitude from each latitude bin is the middle value of the latitude range. 2) The bias can be reduced by using finer latitude bins (e.g. 5°).

L. 150: "Furthermore" does not connect well to the previous sentences, since this is a disadvantage of the scaled-flux water tracing method.

Response: Yes, we removed this word.

L. 153: "Finally" does not appear logical here, since there is no connection between this higher-level comparison of both Eulerian approaches to Lagrangian methods. Maybe you could expand a sentence that explains that this is a more general comparison?

Response: Thank you. We changed 'Finally' to 'In addition'.

L. 155, Section 2.3: These thresholds are still extremely small in comparison to numbers that

can be obtained in the real world situations on a daily basis. With typical snow density, these thresholds would translate to a 0.2 mm and 0.02 mm thick snow layer. I understand that it is possible to use such threshold values in a model context, but some statement mentioning the limited transferability of such threshold values to the real world are advised. The need to use differing definitions of heavy and light precipitation days is evidence of this fact. Some models can produce extremely small precipitation rates for extended periods of time, which affects the lower 10th percentile.

Response: We fully agree the limited transferability to observations. We added: "We note that this definition has limited transferability to station observations."

L. 173: "Though it is lower" - rephrase

Response: Changed to "The simulated accumulation is".

L. 175: "This could partly ..." - those are vastly different explanations. What are the concrete consequences of smoothed topography in the model, and how likely is it that the model is more correct than the accumulation reconstruction?

Response: We split it into two sentences. "The relatively coarse (T63) spatial resolution of the simulation might not be adequate to simulate coastal precipitation. Furthermore, the reconstruction of Antarctic accumulation might be affected by limited ice core records and local processes such as melt events in coastal regions (Monaghan et al., 2006)." The smoothed topography in ECHAM6 could have a major impact on coastal precipitation, which need to be quantified with high-resolution modelling. The accumulation reconstruction is obtained with spatial patterns from model output of MERRA2. Coastal melt events, which are not represented in the model, might lead to model-data mismatch as well.

L. 238: "It suggests that..." this sentence does not add new insight, and is in addition vague. Can something more firmly be extracted from this analysis?

Response: Thank you. We removed this statement. Unfortunately, we do not infer more firm insights from this result now.

L. 254: "This narrow range" - can you expand how the narrow range connects to the role of cyclones? Is there a specific reason why you mention both cyclones and storm tracks?

Response: This is the question that we propose to merit further studies. So we changed these sentences to "Further studies are merited to investigate the relationship between this narrow band of source vel10 of annual mean precipitation over Antarctica and extratropical cyclones propagating along the Southern Ocean storm tracks". I mention only extratropical cyclones now.

L. 258 onwards: Firstly - Secondly are quite far apart and thus hard to relate, rephrase
Response: Thanks. We removed these words.

L. 259: unclear, where can this decoupling be seen?
Response: Thank you. We removed this statement as it does not provide valuable information and can be confusing.

L.281: "the modelling results" rephrase such that it becomes clear that you refer to your own modelling results here
Response: Changed to "our water tracing results"

L. 338: It would be good to differentiate here between a model perspective and nature. For example, add to this sentence "in a climate model framework".
Response: Added.

L. 335: The conclusions should also list some limitations of this study and the overall approach. For example, the limitations of model-simulated precipitation in Antarctic, and the rather coarse resolution of the simulation, which affects the realism of cyclones and fronts, maritime air intrusions, and so on. If you think critically, how much can ultimately be learned about the nature of Antarctic precipitation from this new tool?
Response: We stressed these limitations in the final paragraph and pointed to further research directions: "We note that the results presented here are based solely on a single model simulation. To explore the model dependence of the results, we are developing similar water tracing diagnostics in another atmospheric GCM, the UK Met Office Unified Model. As the coarse spatial resolution of the ECHAM6 T63 simulation might be insufficient to resolve coastal atmospheric flows and marine air intrusions, high-resolution simulations will be conducted in our future studies."
Regarding how much we can ultimately learn from this new tool for Antarctic precipitation, we firstly need to discuss what we want to know about Antarctic precipitation ultimately. The main scientific questions around Antarctic precipitation are its spatial-temporal distribution, temporal variability across multiple scales, what are the driving factors, and how it changes under climate change. These questions are hard to investigate and fully understand with limited observations under harsh environmental conditions. Climate models are necessary, but they might not give the right results because of poor representation of key processes, e.g. Southen Ocean clouds, Antarctic ice sheet surface moisture fluxes. Limited observations might also not be enough to constrain the models physics and parameterizations. Therefore, such a diagnostic tool could be helpful to understand model behaviour, improve the models, and maybe also improve the interpretation of observations, e.g. water isotopes. We have no

doubt that the water tracers are a valuable diagnostic modelling tool, and we consider the full potential of the tools need to be explored with inter-model comparisons, high-resolution modelling, and model-data comparisons.

L. 361: The wind may be linked to cyclones, but also to other features, and simply pressure gradients. Does the wind only play a emphasized role in the evaporation flux variability because of the absence of land regions which could lead to variations in RH?

Response: Thank you for these useful points. It is true that the wind speed can be simply related to pressure gradients. Our results only indicate that given other conditions, higher wind speed favors moisture availability for Antarctic precipitation at individual oceanic grid cells in ECHAM6. Regarding RH or moisture gradient near the ocean surface, we will do such analysis since we can also trace RH related to SST now. However, it remains a question how much it helps to understand the real-world water cycle.

L. 369: I suggest the last paragraph be rewritten to improve the flow and connection between these sentences. Right now they appear merely as an unsorted list.

Response: Thanks. We rewrite it as below:

"We have identified several directions for future research. We note that the results presented here are based solely on a single model simulation. To explore the model dependence of the results, we are developing similar water tracing diagnostics in another atmospheric GCM, the UK Met Office Unified Model. As the coarse spatial resolution of the ECHAM6 T63 simulation might be insufficient to resolve coastal atmospheric flows and marine air intrusions, high-resolution simulations will be conducted in our future studies. While this study focuses on preindustrial conditions, moisture source changes in historical periods, paleoclimate, and future scenarios could also be investigated. Finally, we note that the scaled-flux water tracing approach is applicable not only to Antarctic problems, but also to a range of questions associated with water cycle changes in the rapidly changing environment."